# CURRICULUM LEARNING: A REGULARIZATION METHOD FOR EFFICIENT AND STABLE BILLION-SCALE GPT MODEL PRE-TRAINING

## ABSTRACT

Recent works have demonstrated great success in training high-capacity autoregressive language models (GPT, GPT-2, GPT-3) on a huge amount of unlabeled text corpus for text generation. Despite showing great results, autoregressive models are facing a growing training instability issue. Our study on GPT-2 models (117M and 1.5B parameters) show that larger model sizes, sequence lengths, batch sizes, and learning rates would lead to lower training stability and increasing divergence risks. To avoid divergence and achieve better generalization performance, one has to train with smaller batch sizes and learning rates, which leads to worse training efficiency and longer training time. To overcome this stability-efficiency dilemma, we present a study of a curriculum learning-based approach, which improves the pre-training convergence speed of autoregressive models. More importantly, we find that curriculum learning, as a regularization method, exerts a gradient variance reduction effect and enables to train autoregressive models with much larger batch sizes and learning rates without training instability, further improving the training speed. Our evaluations demonstrate that curriculum learning enables training GPT-2 models with 8x larger batch size and 4x larger learning rate, whereas the baseline approach struggles with training divergence. To achieve the same validation perplexity targets during pre-training, curriculum learning reduces the required number of tokens and wall clock time by up to 61% and 49%, respectively. To achieve the same or better zero-shot WikiText-103/LAMBADA evaluation results at the end of pre-training, curriculum learning reduces the required number of tokens and wall clock time by up to 54% and 70%, respectively.

## 1 INTRODUCTION

Large-scale Transformer-based language models have powered breakthroughs in many natural language processing tasks (Vaswani et al., 2017). These models contain up to billion-scale parameters and are trained on massive open-domain web text corpus with unsupervised learning. Among them, one of the most successful cases is the GPT family: from GPT (110M parameters) (Radford et al., 2018a), to GPT-2 (1.5B parameters) (Radford et al., 2018b), and to GPT-3 with 175B parameters (Brown et al., 2020) that hit record high accuracy for many NLP tasks. Recent studies (Kaplan et al., 2020) show that these models continue to achieve better accuracy as their sizes increase (together with richer training data), indicating the model size will continue growing in the future.

Despite achieving remarkable model accuracy, training these models raise huge challenges on training efficiency and instability. On one hand, training large models require a huge amount of computation. As an example, it requires approximately 9.2 days on 512 V100 GPUs to train a 8.3B GPT-2 (Shoeybi et al., 2019), and 14.8 days on 10000 V100 GPUs to train a 175B GPT-3 (Patterson et al., 2021). To reduce the training wall clock time of these large models, one of the most common solutions is to employ distributed training with hundreds or even thousands of GPUs across multiple machines such that the model can process the training data with much higher throughput. Meanwhile, to avoid the huge overhead incurred by cross-GPU/node communication in distributed training, practitioners increase the batch sizes in the optimization algorithm to increase the computation-communication ratio, to improve training efficiency and reduce training wall clock time.

Despite increased training throughput and efficiency, increasing the batch size is not always the panacea, as it often leads to training instability during pre-training GPT models — training would diverge or recover from divergence but overall lead to slow/worse convergence. We conduct a thorough study of the GPT-2 pre-training task (Radford et al., 2018b; Shoeybi et al., 2019) with different batch sizes, models sizes, sequence lengths, and learning rates. We find that larger batch sizes and longer sequence lengths reduce training stability and increase divergence risk, especially when combined with larger model sizes and learning rate. This often leads to slower and worse convergence that hurts final generalization performance. To achieve stable training and avoid sub-optimal generalization performance, common practice often resorts to using smaller batch sizes and learning rates, which would adversely affect training efficiency.

In this work, we explore a curriculum learning-based approach to address the above training efficiency-stability dilemma. Curriculum learning (CL) was proposed to improve training convergence speed by presenting easier/simpler examples earlier during training and gradually increasing the sample difficulties (Elman, 1993; Sanger, 1994; Bengio et al., 2009). CL was explored and verified for NLP one-stage and fine-tuning tasks (Kocmi & Bojar, 2017; Bojar et al., 2017; Zhang et al., 2018; Platanios et al., 2019; Zhang et al., 2019; Sachan & Xing, 2016; 2018; Tay et al., 2019; Xu et al., 2020), but its application to pre-training GPT models is not well studied. We find that curriculum learning, as expected, could improve the convergence speed under the same training hyperparameters. More importantly, we find that curriculum learning, as a regularization method, can enable stable and efficient training with much larger batch sizes and learning rates than existing baseline approaches. Our analysis shows that one reason for curriculum learning's better training stability lies in its gradient variance reduction effect: Although the gradient variance is generally reduced under larger batch size, the largest variance on certain dimensions (i.e., the outliers) is increased and it reaches extreme values when training divergence happens. Using Adam optimizer's variance term, we observe that curriculum learning helps reduce both the norm of this variance and the maximum variance outliers. This variance reduction effect helps curriculum learning to achieve stable large-batch training without hurting the token-wise convergence speed and final generalization performance.

We implement curriculum learning using sequence length as a primary difficulty metric and apply it to the GPT-2 pre-training with up to 1.5B parameters. Evaluations show that curriculum learning enables stable and efficient training with 8x larger batch size and 4x larger learning rate, where the baseline approach struggles with training divergence. To achieve the same validation perplexity targets during pre-training, curriculum learning reduces the required number of tokens and wall clock time by up to 61% and 49%, respectively. To achieve the same or better zero-shot WikiText-103/LAMBADA evaluation results at the end of pre-training, curriculum learning reduces the required number of tokens and wall clock time by up to 54% and 70%, respectively.

We make the following contributions: i) We conduct an extensive study of the GPT-2 pre-training task, which provides detailed insights about the training stability-efficiency dilemma that motivate our work (Section 3). ii) We present an implementation of curriculum learning based on sequence length for GPT-2 model (and autoregressive model in general), which is both efficient and easy to integrate (Section 4). iii) We conduct large-scale experiments to demonstrate the proposed work's ability to provide superior training stability and efficiency at the same time. To the best of our knowledge, we are the first work to demonstrate the benefit of curriculum learning as a regularization method that improves training stability (Section 5). iv) The curriculum learning implementation as well as the necessary changes to the GPT-2 pre-training framework has been open sourced in a deep learning optimization library (name hidden to maintain anonymity).

## 2 BACKGROUND AND RELATED WORK

**Language model pre-training**    The accuracy of transformer-based language models grows substantially with its model size (Radford et al., 2018a;b; Brown et al., 2020). Today, a large language model such as GPT-3 (Brown et al., 2020) contains up to 175B parameters, and recent studies show that model accuracy can continue to improve with even larger model sizes (Kaplan et al., 2020). However, training these large models often incurs excessively long training time and training difficulties (Brown et al., 2020). Therefore, there are a lot of demands of performing efficient and stable training for large-scale LMs. To have the pre-training finished in a reasonable amount of time, the most common way is to leverage data parallelism to train models on multiple GPUs. However, the speedup gains often saturate beyond a few tens of GPUs, because communication becomes the major bottleneck,

i.e., the workers will spend more time communicating gradients than computing them, as the number of GPUs increases. To mitigate this bottleneck, recent works such as 1-bit Adam (Tang et al., 2021) have studied gradient compression and demonstrate their effectiveness against auto-encoding models such as BERT (Devlin et al., 2019). An alternative approach to alleviate these overheads is to use large batch sizes. For example, LAMB (You et al., 2020) and 1-bit LAMB (Li et al., 2021) enable stable and efficient distributed BERT pre-training with batch size up to 64K/32K (for sequence length 128/512, i.e., 8M/16M tokens per batch) while maintaining the sample-wise convergence speed. For encoder-decoder models such as T5, Raffel et al. (2020) use batch size up to 2K (for sequence length 512, i.e., 1M tokens per batch). For autoregressive models such as the GPT family (Radford et al., 2018a;b; Brown et al., 2020), existing works use batch size up to 1.6K (for sequence length 2K, i.e, 3.2M tokens per batch). Despite the benefit of reduced communication overhead, large-batch training is sensitive to hyperparameters and often leads to issues such as slow convergence, training instabilities, and model divergence (You et al., 2020; Li et al., 2021).

**Curriculum learning**   Inspired by how humans and animals are trained, curriculum learning aims to improve machine learning model training convergence speed by presenting easier (or less complex) examples earlier during training (Elman, 1993; Sanger, 1994; Bengio et al., 2009). Previous studies have demonstrated the benefit of faster convergence speed by curriculum learning in many domains such as natural language processing, computer vision, and neural evolutionary computing. In this section we will discuss recent related works in the natural language processing area. For a more complete literature review, we recommend these recent survey papers (Wang et al., 2020; Soviany et al., 2021).

In the natural language processing area, most of the curriculum learning works focus on small-scale one-stage tasks and downstream fine-tuning tasks, such as neural machine translation (NMT) (Kocmi & Bojar, 2017; Bojar et al., 2017; Zhang et al., 2018; Platanios et al., 2019; Zhang et al., 2019) and natural language understanding (NLU) (Sachan & Xing, 2016; 2018; Tay et al., 2019; Xu et al., 2020). These works show that curriculum learning can improve convergence speed, reduce training time, and improve accuracy. In these works, the curriculum difficulty metrics for each training sample are usually defined as the sentence length, vocabulary frequency, the inference loss on smaller/cheaper models, or based on self-paced learning (Kumar et al., 2010). For the pacing function (i.e., to decide the curriculum difficulty range when sampling next training data batch), these works usually use fixed predefined functions (e.g., gradually increase difficulty upper bound by linear, root, and exponential functions), bucketing heuristics (group data with similar difficulties, and sample from a subset of buckets every time), or based on self-paced learning.

Only a few works explore curriculum learning for language model pre-training. Press et al. (2020) apply curriculum learning to neural language modeling pre-training, specifically a transformer model with 247M parameters (Baevski & Auli, 2018). They find that by adding an additional first training stage with a shorter sequence length, it is possible to achieve the same dev. set perplexity with shorter total training time. Zhang et al. (2021) apply curriculum learning to BERT-base pre-training, a transformer model with 110M parameters. They find that by grouping sequences with similar length and with curriculum learning, it is possible to achieve similar downstream task accuracy with shorter pre-training time. Campos (2021) apply curriculum learning to ELMo pre-training, a bi-directional LSTM model with 93.6M parameters (Peters et al., 2018). They test a variety of curricula on the WikiText-2 and WikiText-103 but do not find strong evidence that the use of curriculum learning can improve language model pre-training.

## 3    MOTIVATION AND INSIGHTS

In this section we perform an in-depth analysis of the GPT-2 model pre-training baseline (without curriculum learning). We follow the training pipeline from the NVIDIA Megatron-LM work (Shoeybi et al., 2019)[1]. All of the experiments are performed on 128 NVIDIA V100 GPUs (32GB memory). There are 16 nodes, and each node has 8 GPUs. GPUs inside the same node are connected by NVLink 2.0, and nodes are connected by a 100 Gigabit InfiniBand EDR inter-node network.

**Model and dataset**   We evaluate two GPT-2 model sizes from the original GPT-2 work (Radford et al., 2018b): one with 117M parameters (12 layers, 768 hidden size, 12 attention heads), and

---

[1]https://github.com/NVIDIA/Megatron-LM

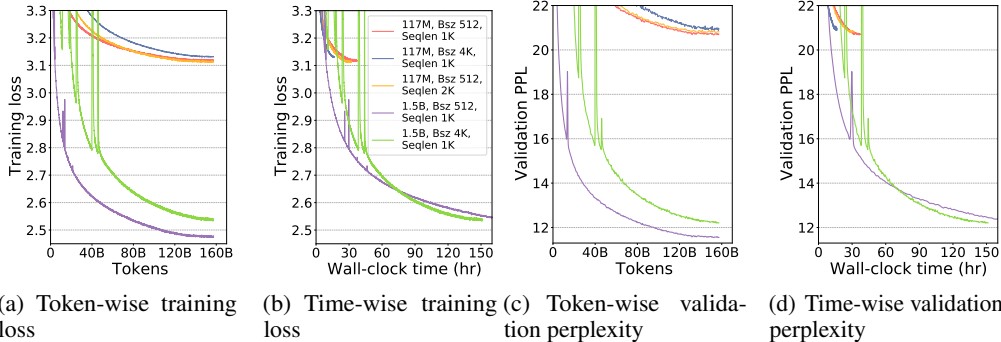

(a) Token-wise training loss

(b) Time-wise training loss

(c) Token-wise validation perplexity

(d) Time-wise validation perplexity

Figure 1: Training loss and validation perplexity during GPT-2 pre-training (baseline without curriculum learning) under different model sizes, batch sizes (and LR), and sequence lengths. All subfigures share the same legend as 1(b). In 1(b) and 1(d), the purple line for "1.5B, Bsz 512, Seqlen 1K" is truncated (the whole training takes 341 hours).

one with 1.5B parameters (48 layers, 1600 hidden size, 25 attention heads). For training data, we collect and use the same dataset blend as the Megatron-LM work: Wikipedia (Devlin et al., 2019), CC-Stories (Trinh & Le, 2018), RealNews (Zellers et al., 2019), and OpenWebText (Radford et al., 2019).

**Training parameters** We evaluate two sets of training parameters. The first set follows the Megatron-LM work: batch size 512, 300K total training steps (157B tokens), and learning rate $1.5 \times 10^{-4}$ with a linear warmup of 3K steps and a single cycle cosine decay over the remaining 297K steps (with minimum learning rate $1 \times 10^{-5}$). The second parameter set tests a more aggressive training strategy: batch size 4K ($8\times$ larger), 37.5K total training steps (157B tokens), and learning rate $6 \times 10^{-4}$ ($4\times$ larger) with a linear warmup of 3K steps and a single cycle cosine decay over the remaining 34.5K steps (same minimum learning rate). For sequence length/context size, we mainly use 1K which is the default for GPT-2. But we also test 2K (on the smaller 117M model with batch size 512 and 157B tokens) which is the default for GPT-3 (Brown et al., 2020). All experiments are performed with mixed precision/FP16 training, Adam optimizer ($\beta_1 = 0.9$, $\beta_2 = 0.999$, $\epsilon = 1 \times 10^{-8}$) (Kingma & Ba, 2015), weight decay of 0.01, checkpoint activation, same random seed (for Python, NumPy, PyTorch, and CUDA), and gradient clipping.

**The stability-efficiency dilemma** Figure 1 presents the training loss and validation perplexity for 5 cases (baseline without curriculum learning) under different model sizes, batch sizes (and LR), and sequence lengths. First, the 1.5B model training is much more unstable and has many divergences. This is even worse when applied with a larger batch size/learning rate, where the final validation perplexity becomes worse as shown in Figure 1(c). Similarly for the 117M model, larger batch size/LR or longer 2K sequence length both lead to worse final validation perplexity (and worse zero-shot WikiText-103/LAMBADA evaluation results as we will later see in Table 2 Section 5).

From the token-wise convergence, it seems that a smaller batch size/learning rate is preferable because of its better generalization performance. However, Figure 1(b) and 1(d) show that such cases would lead to much worse training efficiency. With a larger batch size/learning rate, it is possible to complete the same 157B-token training with much less wall clock time. For the 1.5B model, it takes 341 and 151 hours to complete the training under batch size 512 and 4K, respectively. Overall, this demonstrates the stability-efficiency dilemma for baseline pre-training: With smaller batch size/learning rates, the training is more stable and can achieve better final generalization, but presumably with poorer training efficiency; With larger batch size/learning rates, the training is more efficient, but with lower stability and worse generalization. As a solution, in the next section we will describe how we design and implement curriculum learning, which resolves this dilemma for the GPT-2 model pre-training task.

## 4 DESIGN AND IMPLEMENTATION

**Difficulty metric: sequence length** After considering various metrics used in related works, we choose the sequence length as the curriculum difficulty metric (start from shorter sequence training

data, then gradually increase the sequence length) for two reasons: i) Sequence length as the curriculum difficulty metric has been proven to be effective in the NLP area; ii) For each Transformer block, the self-attention and intermediate layers have time complexity of $O(B \times L^2 \times H)$ and $O(B \times L \times H^2)$, respectively[2]. By reducing the sequence length, we reduce the time complexity quadratically for the self-attention sub-layer and linearly for the intermediate sub-layer of Transformer blocks. To support curriculum learning, traditional implementation would require "training data reordering" to sample from data subsets that are below certain difficulty levels. This adds additional cost before training (to compute the difficulty of data) and during training (data subset sampling). For sequence length metric, we develop a truncation-based implementation that incurs less cost: During the baseline GPT-2 pre-training, the raw text inputs are truncated into sequences with the same length to form a mini-batch regardless of the lengths of the actual sentences. To enable curriculum learning, we still let the dataloader truncate the raw text into samples with the same full sequence length. Then at each step given the sequence length determined by the pacing function, we truncate sequences to form a modified mini-batch for training.

**Pacing function: step-wise linear**   We choose a step-wise linear pacing function with the following definition: Given a starting sequence length $seqlen_1$, an ending sequence length $seqlen_2$ (baseline full sequence length), and a curriculum learning duration $T$ (number of steps), the sequence length used for the training batch at step $t$ is $seqlen_t = seqlen_1 + (seqlen_2 - seqlen_1) \times min(\frac{t}{T}, 1)$. We choose this pacing function for two reasons: First, such fixed heuristic pacing functions are easy to implement and proved effective in the NLP area. Second, step-wise linear function conveys two different functions that are intuitively convincing: i) step-wise linear means that we will perform optimization and update the model for the same number of times for each distinct sequence length; ii) because different sequence lengths lead to different number of tokens in a batch, at longer sequence lengths (higher difficulty) we will learn from more tokens.

Besides step-wise linear, we also explored 3 other pacing functions: i) We tried a discrete 2-stage pacing function from (Press et al., 2020), but it provides less stability benefit (Section 5.4). ii) We tried a step-wise root function ($seqlen_t = seqlen_1 + (seqlen_2 - seqlen_1) \times min((\frac{t}{T})^r, 1)$, where $r$ is the root degree), which performs similar to linear and requires one extra hyperparameter. iii) We tried an adaptive pacing function based on the progress of training/validation loss, which also performs similar and requires extra tuning.

## 5   EVALUATION

In this section we evaluate curriculum learning for GPT-2 pre-training. Our analysis shows that curriculum learning can provide stable training at batch size 4K where baseline struggles with divergence issue. In addition, curriculum learning greatly reduces the required number of tokens and training time to reach the same validation perplexity and zero-shot evaluation results.

**Methodology**   For model, dataset, and hardware, we follow the same methodology in Section 3. For the curriculum learning configurations (defined in Section 4), we use $seqlen_1 = 8/64$ (for 117M/1.5B model) and $seqlen_2 = 1K$ (i.e., the full sequence length). To fully utilize the Tensor Core acceleration in NVIDIA V100 GPU, we add a $seqlen_t = seqlen_t - (seqlen_t \bmod 8)$ postprocessing to make sure the sequence length is always a multiple of 8. For the curriculum duration $T$, we tune this parameter and use different numbers for each case. We analyze and find a low-cost tuning strategy that only requires running the first tens of steps (details in Appendix A.1). For the training parameters, for curriculum learning we use the same shared parameters as the baseline in Section 3 except two parameters: 1) Because during curriculum learning the number of tokens in a data batch is smaller, we modify the training termination condition so that all cases stop when reaching the same 157B training tokens. 2) Because of 1), curriculum learning cases now have more training steps, which make it necessary to modify the learning rate decay schedule to have a fair comparison with the baseline. For 117M model cases, we increase the number of learning rate decay steps by half of the curriculum learning duration. However, we find that simply increasing decay steps still leads to sub-optimal learning rate schedule. So for 1.5B model cases, we change the learning rate decay to token-wise (same cosine decay over the 157B tokens) instead of step-wise. We describe the underlying rationale in Appendix A.2.

---

[2]$B, L, H$ represent the batch size, sequence length, hidden size.

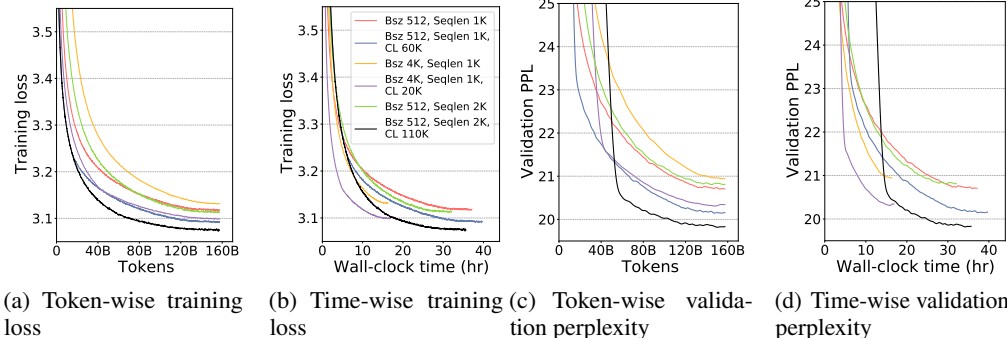

(a) Token-wise training loss    (b) Time-wise training loss    (c) Token-wise validation perplexity    (d) Time-wise validation perplexity

Figure 2: Training loss and validation perplexity during GPT-2 117M pre-training, comparing the baseline and curriculum learning under different batch sizes/LR and sequence lengths. All sub figures share the same legend as 2(b) ("CL 60K" means applying curriculum learning to the first $T$=60K steps).

Table 1: Number of tokens (Billion) and wall clock time (Hour) to reach the same validation perplexity during GPT-2 117M pre-training, comparing the baseline and curriculum learning under different batch sizes/LR and sequence lengths. Last two rows use the best PPL achieved by baseline and CL as the target.

| Vali PPL Target | Bsz 512, Seqlen 1K Baseline / CL 60K Num. tokens / Wall time | Bsz 4K, Seqlen 1K Baseline / CL 20K Num. tokens / Wall time | Bsz 512, Seqlen 2K Baseline / CL 110K Num. tokens / Wall time |
|---|---|---|---|
| 21.5 | 69.0B / 35.55B (-48%) 16.28Hr / 10.81Hr (-34%) | 99.41B / 46.98B (-53%) 10.31Hr / 5.4Hr (-48%) | 73.4B / 51.56B (-30%) 14.96Hr / 13.94Hr (-7%) |
| 21 | 96.78B / 58.51B (-40%) 22.84Hr / 16.25Hr (-29%) | 132.12B / 69.63B (-47%) 13.71Hr / 7.75Hr (-43%) | 102.55B / 53.36B (-48%) 20.9Hr / 14.32Hr (-31%) |
| Baseline best | Target Vali PPL = 20.38 135.63B / 84.46B (-38%) 32.0Hr / 22.4Hr (-30%) | Target Vali PPL = 20.87 156.87B / 77.6B (-51%) 16.27Hr / 8.57Hr (-47%) | Target Vali PPL = 20.55 143.34B / 55.7B (**-61%**) 29.23Hr / 14.82Hr (**-49%**) |
| CL best | Target Vali PPL = **19.85** Did not reach / 136.47B Did not reach / 34.73Hr | Target Vali PPL = **20.23** Did not reach / 146.8B Did not reach / 15.75Hr | Target Vali PPL = **19.6** Did not reach / 135.58B Did not reach / 31.18Hr |

## 5.1 GPT-2 117M EVALUATIONS

**Convergence speed.** Figure 2 demonstrates that curriculum learning provides better token and time-wise convergence compared to the baseline under the same batch size/LR and sequence length. In addition, "Bsz 4K, Seqlen 1K, CL 20K" provides better token and time-wise convergence than "Bsz 512, Seqlen 1K, Baseline", which demonstrates that curriculum learning provides stable and efficient large-batch training while maintaining the convergence speed and final generalization performance. As a result, curriculum learning combined with large-batch training provides a solution to the training stability-efficiency dilemma.

**Tokens/time to reach the same validation perplexity.** To quantitatively measure the token and time-wise convergence speedup, Table 1 summarizes the required number of tokens and training time for baseline and curriculum learning to reach the same validation perplexity during pre-training. In this table, we compare each pair of baseline and curriculum learning cases under the same batch size and sequence length. Results show that curriculum learning reduces the required number of tokens and training time by up to 61% and 49%, respectively. In addition, curriculum learning achieves validation perplexity that baseline is unable to achieve during the whole training.

**Zero-shot evaluation.** Last, we perform zero-shot evaluation of the trained models on the WikiText-103 and LAMBADA datasets using the same methodology in the Megatron-LM work (Shoeybi et al., 2019). In Table 2 we compare all cases to the baseline case with batch size 512 and sequence length 1K, the original configuration in Megatron-LM work. For each curriculum learning (CL) case we present two rows: one evaluated at the earliest checkpoint that provides better eval results; the other one evaluated at the end of full training. At batch size 512, CL provides up to 40%/32% token/time

Table 2: Zero-shot evaluation of the trained models on the WikiText-103 and LAMBADA datasets, following the evaluation methodology from (Shoeybi et al., 2019). Case 2 to 9 are compared with case 1, and case 11 to 17 are compared with case 10. Case 16 (Press et al., 2020) and 17 (Brown et al., 2020) are related works.

| Case | Pre-training parameters | Pre-training steps, tokens, time | WikiText perplexity ↓ | LAMBADA accuracy ↑ |
|---|---|---|---|---|
| **117M:** | | | | |
| 1: Baseline | bsz512-seqlen1K | 300K, 157B, 37Hr | 27.78 | 33.19% |
| 2: CL 60K | bsz512-seqlen1K | 210K, 94B (-40%), 25Hr (-32%) | **27.75** | **34.00%** |
| 3: CL 60K | bsz512-seqlen1K | 330K, 157B (→), 40Hr (+8%) | **27.10** | **34.56%** |
| 4: Baseline | bsz4K-seqlen1K | 37.5K, 157B (→), 16Hr (-57%) | 28.09 | 32.54% |
| 5: CL 20K | bsz4K-seqlen1K | 33K, 96B (-39%), 11Hr (**-70%**) | **27.66** | **33.71%** |
| 6: CL 20K | bsz4K-seqlen1K | 47.5K, 157B (→), 17Hr (-54%) | **27.12** | **34.74%** |
| 7: Baseline | bsz512-seqlen2K | 150K, 157B (→), 32Hr (-14%) | 28.19 | 32.99% |
| 8: CL 110K | bsz512-seqlen2K | 125K, 73B (**-54%**), 18Hr (-51%) | **27.18** | **33.81%** |
| 9: CL 110K | bsz512-seqlen2K | 205K, 157B (→), 36Hr (-3%) | **26.43** | **34.02%** |
| **1.5B:** | | | | |
| 10: Baseline | bsz512-seqlen1K | 300K, 157B, 341Hr | 13.89 | 57.29% |
| 11: CL 270K | bsz512-seqlen1K | 360K, 122B (-22%), 286Hr (-16%) | **13.89** | **57.38%** |
| 12: CL 270K | bsz512-seqlen1K | 428K, 157B (→), 364Hr (+7%) | **13.88** | **57.89%** |
| 13: Baseline | bsz4K-seqlen1K | 37.5K, 157B (→), 151Hr (-56%) | 14.76 | 55.06% |
| 14: CL 45K | bsz4K-seqlen1K | 50K, 121B (-23%), 121Hr (-65%) | **13.88** | **58.20%** |
| 15: CL 45K | bsz4K-seqlen1K | 58.8K, 157B (→), 155Hr (-55%) | **13.72** | **58.47%** |
| 16: 2-stage CL 20K | bsz4K-seqlen1K | 55K, 157B (→) | 14.14 | 57.23% |
| 17: Bsz Warmup 45K | bsz4K-seqlen1K | 58.8K, 157B (→) | 14.21 | 56.36% |

reduction while achieving better evaluation results (case 1, 2). After full training, CL takes 8% more time because it needs more steps to reach the same 157B tokens, but more importantly CL provides even better eval results (case 3). When increasing the batch size or sequence length for baseline, the training time is reduced but the evaluation results get worse due to less stable training and worse convergence (case 4, 7). On the other hand, CL with a larger batch size/sequence length can provide both great token/time saving (up to 54%/70%) and better eval results (case 5, 6, 8, 9).

## 5.2 GPT-2 1.5B EVALUATIONS

**Convergence speed.** Figure 3 demonstrates that curriculum learning provides better token-wise convergence compared to the baseline under the same batch size. In addition, "Bsz 4K, CL 15K" provides better token and time-wise convergence than "Bsz 512, Baseline", which demonstrates that curriculum learning provides stable and efficient large-batch training while maintaining the convergence speed. For the batch size 512 case, curriculum learning provides smaller time-wise speedup. This is because in this case, the communication overhead is too high so that the time saving provided by curriculum learning's shorter sequences becomes relatively smaller. Another reason is that the baseline is more stable at batch size 512.

**Tokens/time to reach the same validation perplexity.** Table 3 summarizes the required number of tokens and training time for baseline and curriculum learning to reach the same validation perplexity. Results show that curriculum learning reduces the required number of tokens and training time by up to 45% and 41%, respectively. Compared to baseline with batch size 512, curriculum learning with batch size 4K can achieve better final validation perplexity together with much better training efficiency, demonstrating the dual stability-efficiency capability.

**Zero-shot evaluation.** At batch size 512, curriculum learning can achieve better eval results with 22%/16% less token/time and achieve even better eval results after full training (case 10, 11, 12 in Table 2). Baseline with a larger batch size 4K has better efficiency but worse stability/eval results (case 13), while CL with a larger batch size provides both efficiency and stability (case 14, 15).

## 5.3 VARIANCE REDUCTION HELPS STABILIZE TRAINING.

For stochastic gradient optimization, when the gradient variance is large, the algorithm might spend much time bouncing around, leading to slower convergence and potential divergence (Wang et al.,

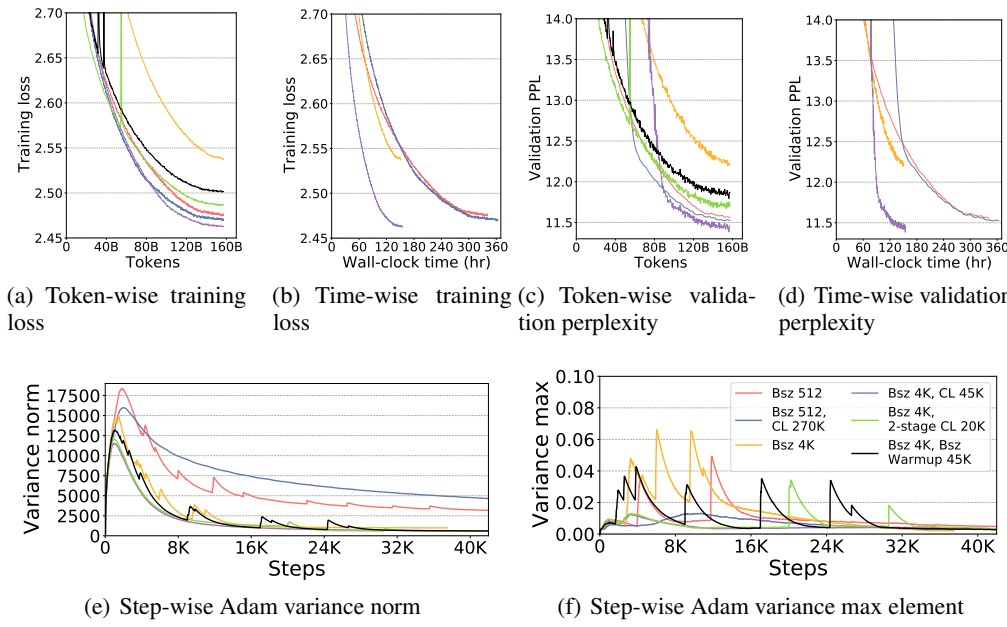

(a) Token-wise training loss

(b) Time-wise training loss

(c) Token-wise valida-tion perplexity

(d) Time-wise validation perplexity

(e) Step-wise Adam variance norm

(f) Step-wise Adam variance max element

Figure 3: Training loss, validation perplexity, and Adam variance norm/max element during GPT-2 1.5B Seqlen 1K pre-training, comparing the baseline and curriculum learning under different batch sizes/LR. Also compare with two related works ("2-stage CL" (Press et al., 2020) and "Bsz Warmup" (Brown et al., 2020)) described in Section 5.4. These two related works are not presented in 3(b) and 3(d) because they are performed on different hardware. All subfigures share the same legend as 3(f) ("CL 45K" means applying curriculum learning to the first $T$=45K steps).

Table 3: Number of tokens (Billion) and wall clock time (Hour) to reach the same validation perplexity during GPT-2 1.5B Seqlen 1K pre-training, comparing the baseline and curriculum learning under different batch sizes/LR. Last two rows use the best PPL achieved by baseline and CL as the target.

| Vali | Bsz 512 | Bsz 4K |
|---|---|---|
| PPL | Baseline / CL 270K | Baseline / CL 45K |
| Target | Num. tokens / Wall time | Num. tokens / Wall time |
| 12.25 | 74.24B / 61.64B (17% ↓) | 148.48B / 85.45B (42% ↓) |
| | 161.38Hr / 153.28Hr (5% ↓) | 142.86Hr / 87.76Hr (39% ↓) |
| 12 | 83.62B / 68.87B (18% ↓) | Did not reach / 87.79B |
| | 181.71Hr / 169.44Hr (7% ↓) | Did not reach / 89.99Hr |
| Baseline best | Target Vali PPL = 11.4 | Target Vali PPL = 12.2 |
| | 155.29B / 143.91B (7% ↓) | 154.77B / 85.45B (45% ↓) |
| | 336.74Hr / 334.23Hr (1% ↓) | 148.91Hr / 87.76Hr (41% ↓) |
| CL best | Target Vali PPL = **11.37** | Target Vali PPL = **11.38** |
| | Did not reach / 152.56B | Did not reach / 156.16B |
| | Did not reach / 353.21Hr | Did not reach / 154.29Hr |

2013). Previous studies show that variance reduction methods improve training stability in areas such as reinforcement learning (Mao et al., 2018; Cheng et al., 2019; Anschel et al., 2017). During our study of curriculum learning, one surprising finding is that curriculum learning can, in addition to the well-known benefit of faster convergence speed, help to reduce this gradient variance norm/max element, which is presumably why it leads to much more stable training under large model size and large batch size/LR. Figure 3(e) and 3(f) plot the $l_1$ norm[3] and max element of Adam's variance term ($\sqrt{v_t}$, where $v_t = \beta_2 v_{t-1} + (1 - \beta_2)(g_t)^2$) for the GPT-2 1.5B case. We plot them step-wise since the optimization and model update are performed at every step.

We see that when batch size/LR increases, the variance norm decreases but the max element increases. When comparing GPT-2 117M and 1.5B cases (not shown in figure), we see that when model size increases both the variance norm and max element increase. When sequence length increases for

---

[3]Here we use $l_1$ norm to avoid outlier amplification.

the GPT-2 117M case, the variance norm stays the same but the max element increases. These demonstrate that large gradient variance norm/max element is one of the symptoms of the training instability issue, similar to what was observed in the aforementioned studies in reinforcement learning area. We find that curriculum learning stabilizes training and reduces both the Adam variance norm and the variance max element. Importantly, curriculum learning avoids all the spikes of the variance max element, which all happen to be where the baseline training diverges. This is one of the reasons why curriculum learning can provide substantial token/time-wise speedup for large-batch training. One may wonder why gradient clipping cannot help avoid these extreme gradient variance outliers. Although gradient clipping can avoid too large/small gradient at every single step, it cannot avoid the gradient variance getting accumulated at certain dimensions, especially for large batch sizes.

### 5.4 COMPARING WITH RELATED WORKS

In the end, we compare the proposed work with two related works on the "1.5B model with batch size 4K" case which is the most challenging one. The first work is a "2-stage curriculum learning" where the first stage uses a shorter sequence length and the second stage uses the full sequence length (Press et al., 2020). Following the tuning strategy in the related work, we set the sequence length as 128 for the first stage and set its duration at about half of the baseline duration (20K steps). The second work is the "batch size warmup" technique used by GPT-3 pre-training where the training starts with a smaller batch size (sequence length unchanged) and gradually increases to full batch size (Brown et al., 2020). In our experiment, we set the starting batch size at 128 and then gradually increase it to 4K, and set the warmup duration at 45K steps (same as the proposed work). Both related works use the same training parameters as the proposed work. Because we perform the two experiments on a different cluster, we do not report the training time as it is not comparable.

Figure 3 compares the training loss, validation perplexity, and Adam variance norm/max element. Both related works provide convergence speedup but it is less than the proposed work. More importantly, both related works still have training instability issues. The two-stage curriculum learning has an obvious training divergence at step 20K when the sequence length switches from 128 to 1K (the spike at 20K in Figure 3(f)). This is because when staying at the same shorter sequence length for too long, the model becomes heavily overfitted for that length which leads to divergence risk when switching to full length. Although both batch size warmup and curriculum learning reduce the number of tokens per batch in a similar fashion, batch size warmup does not provide any training stability benefit compared to the baseline. This indicates that providing the same number of shorter (less complex) sequences leads to better training stability than providing fewer number of same length (same complexity) sequences. In addition, batch size warmup has a limitation that the batch size must be multiple of data-parallel size, which will be large for distributed training. On the other hand, for curriculum learning the sequence length only needs to be multiple of 8 to enable Tensor Core acceleration. Both related works provide worse zero-shot evaluation results (case 16, 17 in Table 2).

## 6 CONCLUSION AND DISCUSSION

It is well known that curriculum learning, by presenting easier or simpler training data first, could provide faster convergence speed under the same training parameters. On the other hand, our study reveals another important benefit of curriculum learning as a regularization method: enable stable training at larger batch size and learning rate by providing a gradient variance reduction effect. Our evaluations on GPT-2 pre-training with up to 1.5B parameters demonstrate that by enabling stable and efficient training with 8x batch size and 4x learning rate, curriculum learning can reduce up to 54%/70% training token/time while still achieving better zero-shot evaluation results.

We hope that our work could motivate more studies of curriculum learning as a regularization method and its applications in other cases such as autoencoding language models and even computer vision models. We believe with a proper curriculum learning design, it is possible to observe similar training stability/efficiency benefits by curriculum learning in other areas. We also hope that our work could motivate more studies of training data efficiency in general: as training data keep accumulating, sooner or later one will not be able to finish even one epoch for many training tasks. In this situation, intelligent data pipeline techniques (such as but not limited to curriculum learning) would become increasingly critical to the training stability/efficiency.

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

Table 4: Zero-shot evaluation of the trained 117M models on the WikiText-103 and LAMBADA datasets, following the evaluation methodology from (Shoeybi et al., 2019).

| Case | Pre-training parameters | Pre-training steps, tokens, time | Pre-training test perplexity ↓ | WikiText-103 perplexity ↓ | LAMBADA accuracy ↑ |
|------|------|------|------|------|------|
| 1: Baseline | bsz512-seqlen1K | 300K, 157B, 37Hr | 20.75 | 27.78 | 33.19% |
| 2: CL 20K | bsz512-seqlen1K | 310K, 157B, 38Hr | 20.52 | 27.41 | 34.23% |
| 3: CL 40K | bsz512-seqlen1K | 320K, 157B, 39Hr | 20.22 | **26.94** | 33.69% |
| **4: CL 60K** | bsz512-seqlen1K | 330K, 157B, 40Hr | **20.17** | 27.10 | **34.56%** |
| 5: CL 80K | bsz512-seqlen1K | 340K, 157B, 40Hr | 20.26 | 27.04 | 34.16% |
| 6: CL 100K | bsz512-seqlen1K | 350K, 157B, 41Hr | 20.29 | 27.14 | 34.19% |
| 7: Baseline | bsz4K-seqlen1K | 37.5K, 157B, 16Hr | 20.99 | 28.09 | 32.54% |
| 8: CL 10K | bsz4K-seqlen1K | 42.5K, 157B, 17Hr | 20.37 | 27.20 | 34.19% |
| 9: CL 15K | bsz4K-seqlen1K | 45K, 157B, 17Hr | **20.34** | 27.28 | 34.58% |
| **10: CL 20K** | bsz4K-seqlen1K | 47.5K, 157B, 17Hr | 20.36 | **27.12** | **34.74%** |
| 11: CL 25K | bsz4K-seqlen1K | 50K, 157B, 17Hr | 20.38 | 27.41 | 33.59% |
| 12: CL 30K | bsz4K-seqlen1K | 52.5K, 157B, 17Hr | 20.40 | 27.30 | 33.98% |
| 13: Baseline | bsz512-seqlen2K | 150K, 157B, 32Hr | 20.87 | 28.19 | 32.99% |
| 14: CL 50K | bsz512-seqlen2K | 175K, 157B, 34Hr | 20.17 | 26.59 | 33.48% |
| 15: CL 70K | bsz512-seqlen2K | 185K, 157B, 34Hr | 19.99 | 26.34 | 33.30% |
| 16: CL 90K | bsz512-seqlen2K | 195K, 157B, 35Hr | 19.93 | **26.26** | 32.68% |
| **17: CL 110K** | bsz512-seqlen2K | 205K, 157B, 36Hr | **19.90** | 26.43 | **34.02%** |
| 18: CL 130K | bsz512-seqlen2K | 215K, 157B, 36Hr | 19.94 | 26.48 | 33.24% |

# A  APPENDIX

## A.1  CURRICULUM LEARNING HYPERPARAMETERS TUNING STRATEGY

To find a good tuning strategy for curriculum learning, we first perform a simple grid search on the smaller 117M model. Table 4 summarizes the grid search results for the curriculum learning duration ($T$ in Section 4) where we set the starting sequence length ($seqlen_1$ in Section 4) fixed at 8. We then choose the "best" duration based on the test data perplexity and zero-shot evaluation results at the end of 157B-token pre-training. One thing to note is that all the CL cases have quite comparable zero-shot evaluation results, indicating that CL's performance is not very sensitive to the CL duration within a reasonable range.

On the other hand, the above grid search also sheds light on a better tuning strategy: for all three batch size-sequence length combinations, we find that the "best" curriculum duration is always the longest duration that does not have significant validation perplexity fluctuation during the first 10K steps (i.e., a few multiples of the LR warmup steps). In Figure 4(a) the CL 60K is the longest duration we tested that does not have significant validation fluctuation during the first 10K steps, and in Figure 4(b) and Table 4 CL 60K does provide the best validation perplexity during second half of training, best test perplexity at the end, and best LAMBADA accuracy. Since this does not require training the model until full convergence, this heuristic reduces the hyperparameter tuning cost of the CL-based approach. For the 1.5B model, we used this strategy to select the curriculum learning duration based on the validation perplexity curve at the beginning.

For the starting sequence length $seqlen_1$, our grid search results show that it's generally better to set it as small as possible, to maximize the stability and convergence speedup benefit. However, if the validation perplexity has significant fluctuation at the very beginning, it is better to increase the starting sequence length. This is why we choose 8 and 64 as the starting sequence length for 117M and 1.5B models, respectively. Note that this starting sequence length should be a multiple of 8 to enable Tensor Core acceleration.

Overall, a general tuning strategy can be summarized as:

1) Start with $seqlen_1 = 8$ and $T = $ a few multiples of LR warmup steps.

2) Increase $seqlen_1$ until the validation perplexity no longer has significant fluctuation at the very beginning.

3) Perform a binary search to find the largest $T$ that does not have significant validation perplexity fluctuation during the first few multiples of LR warmup steps.

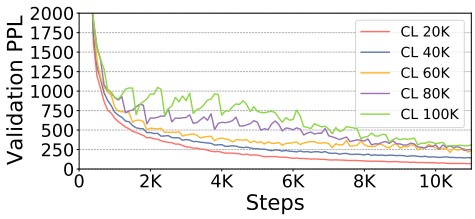 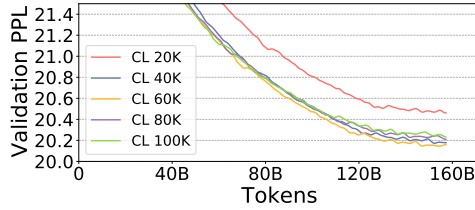

(a) Step-wise validation perplexity (beginning of training)

(b) Token-wise validation perplexity (second part of training)

Figure 4: Step-wise and token-wise validation perplexity during GPT-2 117M Seqlen 1K pre-training with batch size 512 and different curriculum learning duration. ("CL 20K" means applying curriculum learning to the first $T$=20K steps).

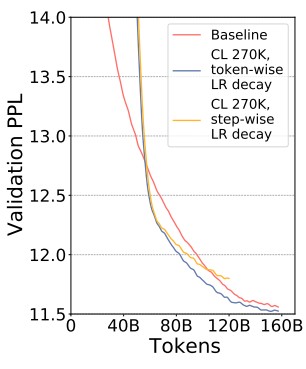 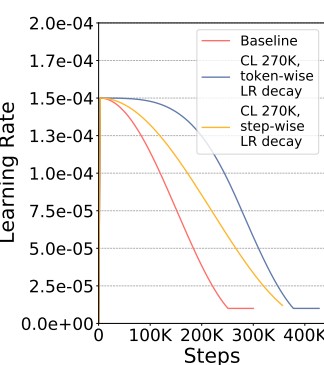 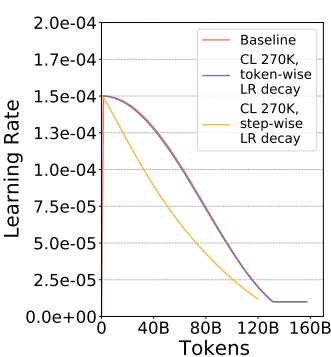

(a) Token-wise validation perplexity

(b) Step-wise learning rate

(c) Token-wise learning rate

Figure 5: Validation perplexity and learning rate during GPT-2 1.5B Seqlen 1K pre-training with batch size 512, comparing the baseline and curriculum learning under different learning rate decay schedules ("CL 270K" means applying curriculum learning to the first $T$=270K steps).

## A.2 LEARNING RATE SCHEDULE FOR CURRICULUM LEARNING

As described in Section 5, for 117M model CL cases we increase the number of learning rate decay steps by half of the curriculum learning duration since CL cases require more steps to reach 157B tokens. However, we find that simply increasing decay steps still leads to sub-optimal learning rate schedule, and for 1.5B model cases we change the learning rate decay to token-wise (same cosine decay over the 157B tokens) instead of step-wise. This is because for CL cases there are fewer tokens per step at the beginning. So even if we increase the LR decay steps, it still cannot avoid decaying faster token-wise at the beginning. As shown in Figure 5, CL with step-wise LR decay has faster token-wise LR decay compared to baseline, which leads to a worse validation perplexity curve. On the other hand, the same CL case with token-wise LR decay has a comparable LR schedule compared to baseline, which leads to better convergence.

## A.3 ADDITIONAL ANALYSIS ABOUT TRAINING STABILITY

This section aims to provide additional information and analysis about training stability. First, because in main paper we need to show the final convergence difference between baseline and curriculum learning, we had to truncate a lot on y axis when plotting. This makes it harder to observe baseline's training instability issue, which mostly happened at the first half of training when the loss is still higher. Figure 6 presents the re-plotting of Figure 2(a) and Figure 3(a) but with less y-axis truncation. At 117M, the baseline has a few training loss spikes at batch size 4K. At 1.5B, the baseline and related works experience significant training instability exhibited as many loss spikes when training with either batch size 512 or 4K. In contrast, curriculum learning leads to a much healthier training curve and shows no loss spikes in the entire training process under both model sizes and batch sizes,

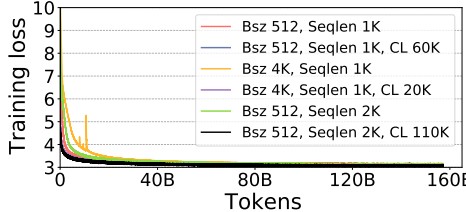 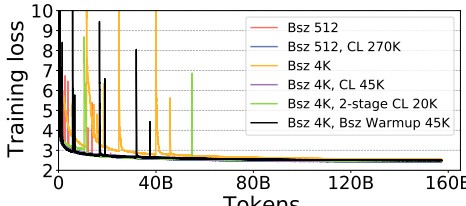

(a) Token-wise training loss for 117M GPT-2. This is the same plot as Figure 2(a) but with less y-axis truncation.

(b) Token-wise training loss for 1.5B GPT-2. This is the same plot as Figure 3(a) but with less y-axis truncation.

Figure 6: Training loss during GPT-2 pre-training, comparing the baseline, related works and curriculum learning.

Table 5: Measuring training instability by the ratio between the current step training loss and the minimum loss among all previous steps. Larger ratios (esp. those greatly larger than 1.0) indicate larger training instability/divergence.

| | | #steps with ratio > 1.2 (% of total steps) | #steps with ratio > 1.5 (% of total steps) | Max ratio |
|---|---|---|---|---|
| **117M:** | | | | |
| 1: Baseline | bsz512-seqlen1K | 0 (0.0%) | 0 (0.0%) | 1.048 |
| 2: CL 60K | bsz512-seqlen1K | 0 (0.0%) | 0 (0.0%) | 1.06 |
| 3: Baseline | bsz4K-seqlen1K | 22 (0.0587%) | 0 (0.0%) | 1.421 |
| 4: CL 20K | bsz4K-seqlen1K | 0 (0.0%) | 0 (0.0%) | 1.016 |
| 5: Baseline | bsz512-seqlen2K | 0 (0.0%) | 0 (0.0%) | 1.045 |
| 6: CL 110K | bsz512-seqlen2K | 0 (0.0%) | 0 (0.0%) | 1.041 |
| **1.5B:** | | | | |
| 7: Baseline | bsz512-seqlen1K | 114 (0.038%) | 36 (0.012%) | 2.101 |
| 8: CL 270K | bsz512-seqlen1K | 0 (0.0%) | 0 (0.0%) | 1.063 |
| 9: Baseline | bsz4K-seqlen1K | 1381 (3.6828%) | 749 (1.9974%) | 5.65 |
| 10: CL 45K | bsz4K-seqlen1K | 0 (0.0%) | 0 (0.0%) | 1.018 |
| 11: 2-stage CL 20K | bsz4K-seqlen1K | 219 (0.3982%) | 184 (0.3346%) | 2.856 |
| 12: Bsz Warmup 45K | bsz4K-seqlen1K | 1179 (2.0062%) | 821 (1.397%) | 4.319 |

and this healthier training curve leads to better convergence and zero-shot evaluation results in main paper Section 5.

To quantitatively study "training instability", we define "loss ratio" as an indicative measurement of training instability, which measures the ratio between the current step training loss and the minimum loss among all previous steps. A ratio larger than 1.0 means that current step's loss is larger than the minimum loss in previous steps, thus larger ratio indicates a larger instability. Table 5 summarizes the number of steps with loss ratio larger than 1.2 and 1.5, and the maximum loss ratio during the training for all baseline and curriculum learning cases reported in the main paper. Similar to what we observe in Figure 6(a), at 117M model size only the baseline with batch size 4K has 22 steps with high loss ratios up to 1.421, and all other cases demonstrate much smoother training loss trend. On the other hand, at 1.5B model size the baseline with both batch size 512 and 4K has many steps with large loss ratios, and with the maximum loss ratio as high as 5.65. Baseline with batch size 4K is less stable than baseline with batch size 512, indicating that larger batch sizes could lead to more training instability risks. In the case of related works, the 2-stage curriculum learning reduces the number and max of large loss ratios but could not fully resolve the instability issue, and the batch size warmup technique provides very minimal training stability gain. Finally, the proposed curriculum learning method has zero steps with large loss ratios under both batch size 512 and 4K, demonstrating its superior training stability.

## A.4 ADDITIONAL ANALYSIS ABOUT TRAINING HYPERPARAMETERS

In Appendix A.1 above we demonstrate that curriculum learning's two hyperparameters can be tuned with very low cost only running the very beginning of the training (the third hyperparameter, ending sequence length, does not require tuning since it will always be the full length). To understand more about how curriculum learning affects the choice and tuning of normal training hyperparameters, this

Table 6: Number of steps with training loss ratios (defined in Appendix A.3) larger than 1.5 during GPT-2 1.5B Seqlen 1K pre-training (first 3K steps only) with batch size 2K, 5 different seeds, and different learning rates for baseline and curriculum learning. Left/right number in each cell is for baseline/CL, respectively.

| Baseline/CL #loss ratio > 1.5 | LR = $1.5 \times 10^{-4}$ | LR = $3 \times 10^{-4}$ | LR = $6 \times 10^{-4}$ | LR = $12 \times 10^{-4}$ |
|---|---|---|---|---|
| Seed 1234 | 0/0 | 296/0 | 359/0 | 179/74 |
| Seed 1235 | 0/0 | 302/0 | 408/0 | 555/459 |
| Seed 1236 | 0/0 | 0/0 | 569/0 | 626/414 |
| Seed 1237 | 7/0 | 0/0 | 548/0 | 614/139 |
| Seed 1238 | 0/0 | 0/0 | 121/0 | 394/29 |
| Total | 7/0 | 598/0 | 2005/0 | 2368/1115 |

section provides additional analysis about learning rates and gradient clipping. Results demonstrate that (a) Compared to baseline, curriculum learning requires less tuning effort on these hyperparameters to provide a stable training; (b) By enabling stable training on larger learning rates, curriculum learning could provide better training efficiency and convergence (as demonstrated in main paper Section 5); (c) Tuning gradient clipping for baseline could not provide the same training stability as curriculum learning.

### A.4.1 LEARNING RATE

In Section 5 we demonstrate that curriculum learning can provide stable and more efficient training at larger batch size and learning rate, where baseline suffers from training instability. We increased both batch size[4] and learning rate at the same time because (a) Large-batch training is more efficient for large-scale training, so larger batch was necessary in our study (b) In order to maintain the same convergence speed, it is necessary to increase the learning rate under larger batch size. A well-known rule of thumb is that the learning rate should at least increase by the square root of the batch size's increase ratio.

As a controlled experiment, here we perform additional studies about what if we keep the batch size the same and only tune learning rate for baseline and curriculum learning. We do not consider the case of "same learning rate, different batch sizes" due to the reason (b) above. Table 6 presents the number of steps with training loss ratios (defined in Appendix A.3 as an indicative measurement of training instability) larger than 1.5 during GPT-2 1.5B Seqlen 1K pre-training (first 3K steps only) with batch size 2K[5], 5 different seeds, and different learning rates for baseline and curriculum learning. And Figure 7 illustrates some of the cases with seed 1236 to show how the loss spikes look like. Results show that curriculum learning provides stable training during this first 3K steps for all five seeds at learning rates up to $6 \times 10^{-4}$, while baseline with seed 1237 still has 7 large loss ratios at learning rate as low as $1.5 \times 10^{-4}$. At learning rate $12 \times 10^{-4}$ both cases have large loss ratios, but curriculum learning reduces the frequency by 2.1x. This demonstrates that (a) Larger learning rates lead to higher training instability risk for both cases. (b) With the same amount of tuning effort, curriculum learning has a higher probability of providing a stable training because of the wider range of learning rates it enables; (c) Since curriculum learning enables stable training at larger learning rate, it could provide better and faster training convergence as shown in main paper Section 5.

### A.4.2 GRADIENT CLIPPING

In main paper Section 5 we used gradient clipping at 1.0 (global gradient $l_2$ norm is clipped to 1.0) following the previous work (Shoeybi et al., 2019). Here we perform additional studies about what if we apply more gradient clipping to baseline. Figure 8(a) presents the training loss during GPT-2 1.5B Seqlen 1K pre-training (first 5K steps only) with batch size 4K (the same hyperparameters as the second set in Section 3), comparing the baseline and curriculum learning under different gradient

---

[4]When batch size increases in Section 5, we also decrease number of training steps because for language model pre-training it is common to keep number of training tokens the same for a fair comparison about zero-shot evaluation. Similarly, under the same batch size the curriculum learning case would have more total training steps since there are less tokens per step at the beginning.

[5]Batch size 2K is used here because this analysis was performed at an early stage of this work, and we did not have enough resource to rerun the same analysis with batch size 4K.

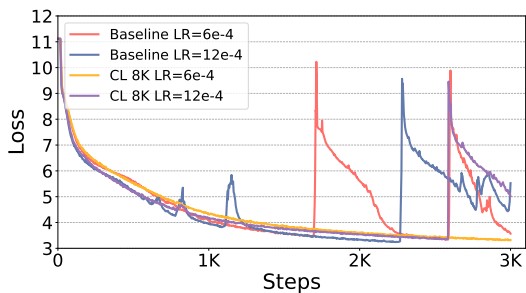

Figure 7: Step-wise training loss during GPT-2 1.5B Seqlen 1K pre-training (first 3K steps only) with batch size 2K, seed 1236, and different learning rates for baseline and curriculum learning ("CL 8K" means applying curriculum learning to the first $T$=8K steps).

clipping levels[6]. Results show that when applying more gradient clipping to baseline, the training has less and smaller loss spikes. And the Adam varaince norm is also reduced as shown in Figure 8(c).

However, more gradient clipping does not fully resolve the training stability issue. Even baseline with the lowest gradient clipping norm cannot avoid all training loss spikes, while curriculum learning with default gradient clipping has no loss spike. As described in main paper end of Section 5.3, we believe that this is a limitation of common gradient clipping technique: Although gradient clipping can avoid too large gradient at every single step, it cannot avoid the gradient variance getting accumulated at certain dimensions (as shown in Figure 8(d)), especially for large batch sizes. Another concern about applying more gradient clipping is that the momentum norm is also reduced due to more clipping (Figure 8(b)). This indicates that when later the training reaches a more stable stage, more gradient clipping could hurt the convergence speed. On the other hand, curriculum learning will not affect the convergence speed after the full sequence length is reached. Another thing to note is that curriculum learning relies less on gradient clipping: at gradient clipping norm 1.0, baseline has 798 clippings in the first 5K steps while curriculum learning has 628 clippings (21% less).

Overall, this analysis demonstrates that curriculum learning requires less or no tuning on gradient clipping, while baseline still has training stability issue with more gradient clipping. It is possible that more complex and adaptive gradient/variance/activation clipping techniques could potentially achieve the same level of training stability as curriculum learning. However, inventing and applying such techniques would require an effort no lower than the proposed curriculum learning solution, which is both easy to integrate and low-cost to tune.

### A.5 LONGER SEQUENCES DO INCUR HIGHER TRAINING INSTABILITY RISKS

To learn more about whether sequence length (as used by many previous works in Section 2) is an appropriate curriculum learning metric, here we study whether longer sequences incur higher training instability risks. We design an experiment where we artificially insert full-length sequences during curriculum learning. Specifically, for every 375 steps, we insert 37 steps of full-length sequences (by disabling the truncation) after 338 steps of normal curriculum learning with short sequence lengths[7]. Figure 9 presents the step-wise training loss for baseline, normal curriculum learning, and the curriculum learning with the artificial 10% of full-length sequences. There are three observations based on the results: (a) After inserting 10% of full-length sequences, curriculum learning starts to have training loss spikes and those spikes tend to happen at where the full-length sequences are inserted, which indicates that longer sequences do incur higher training instability risks; (b) However, the loss spikes are smaller than the baseline's, which indicates that curriculum learning could be more resilient to rare and complex data; (c) The loss spikes happen less frequently from step 4K, which indicates that the training with curriculum learning gets more and more stable.

---

[6]We also tried less than 0.25 gradient clipping, which triggered a silent crash without error messages after around 100 steps. We did not have enough time to find the root cause, but it could be caused by the too extreme gradient clipping.

[7]We choose 375 because in this curriculum learning case we increase the sequence length every 375 steps.

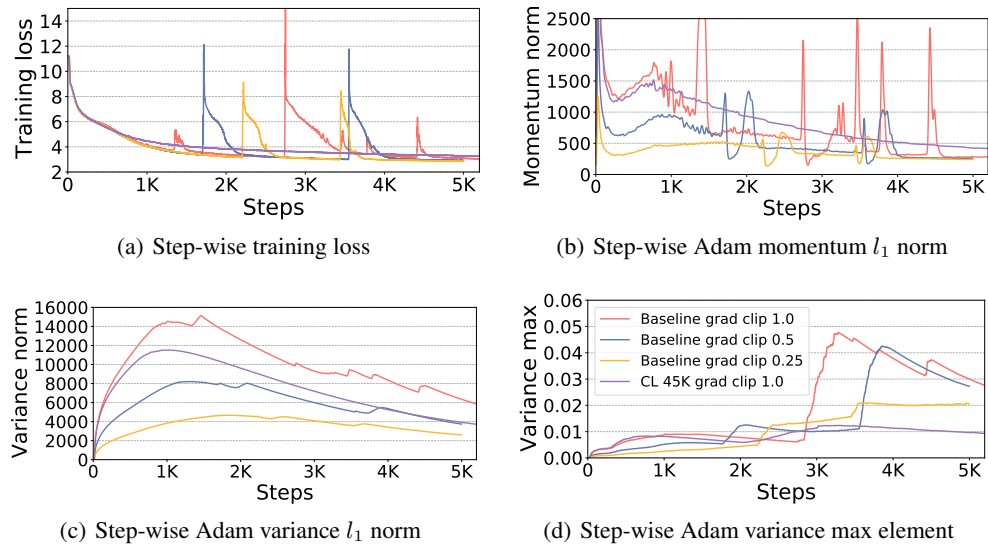

(a) Step-wise training loss

(b) Step-wise Adam momentum $l_1$ norm

(c) Step-wise Adam variance $l_1$ norm

(d) Step-wise Adam variance max element

Figure 8: Training loss, Adam momentum $l_1$ norm, and Adam variance $l_1$ norm/max element during GPT-2 1.5B Seqlen 1K pre-training (first 5K steps only) with batch size 4K, comparing the baseline and curriculum learning under different gradient clipping levels. Grad clip 1.0 indicates that the global gradient $l_2$ norm is clipped to 1.0. All subfigures share the same legend as 8(d) ("CL 45K" means applying curriculum learning to the first $T$=45K steps).

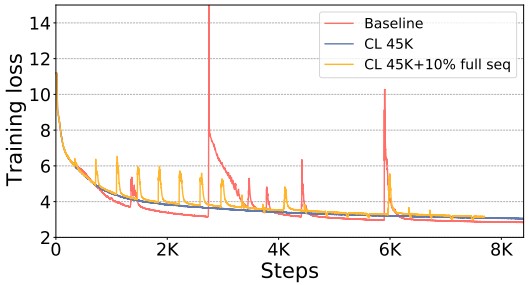

Figure 9: Step-wise training loss during GPT-2 1.5B Seqlen 1K pre-training (first 8K steps only) with batch size 4K (the same hyperparameters as the second set in Section 3), comparing baseline, curriculum learning, and curriculum learning plus inserting 10% of full-length sequences every 375 steps.

