# OpenReview forum: "Curriculum Learning: A Regularization Method for Efficient and Stable Billion-Scale GPT Model Pre-Training"
_ICLR.cc/2022/Conference — ICLR 2022 Submitted_

### Official Review · Reviewer_D2sE · 2021-10-16

**Correctness:** 4
**Technical Novelty And Significance:** 2
**Empirical Novelty And Significance:** 3
**Recommendation:** 5
**Confidence:** 3

**Main Review:**

Strengths
- Method is simple and easy to understand
- Topic of efficient pretraining is important

Weaknesses (reasons to reject)
- In terms of new ideas, the paper is not particularly ambitious or novel. I am not an expert, but given the cited related work, the results of this paper are not surprising or thought-provoking to me. Curriculum learning has been shown to improve training speed in a variety of scenarios, so I am not at all surprised that if one tries hard, they could make it work with GPT pretraining. Sorry if this is not actionable.
- The paper claims that the new lesson learned is that stability is improved with curriculum learning. If a convincing result for this was shown, I have missed it. I think part of it is that it is still unclear to me what the particular operationalization of "stability" is that is used in the paper. The fact it is really hard to match the colors to lines in Figure 3 doesn't help, either. For instance, where is the training instability in Figure 2? Did the GPT-2 authors originally struggle a lot to do "stable" training? If there is prior work on operationalizing stability, then it should probably be explained further in the paper, given that stability is the novel contribution.
- To make the above bullet point even worse, the use of the word "regularization" further convolutes what the main learning is. My understanding of regularization is that it prevents overfitting. Where's the overfitting in these charts?

Weaknesses (not reasons to reject)
- The paper would be more attractive in my opinion if they really made it more about curriculum learning than it currently is. What I mean by this is that the current paper seems more like "BERT trains fast and with more stability if you gradually increase the sequence length of examples." It would be a lot more interesting, for example, if it were shown that the author's proposed method worked for other operationalizations of curriculum learning in addition to sequence length (e.g., age of acquisition, word frequency, etc).
- Related to the above point, there is very little motivation for why sequence length is a correct operationalization.
- In fact, I do not think it is even shown that curriculum learning is the reason for the improvement. For example, the author's experiments do not distinguish curriculum learning from the simple hypothesis "putting similar examples into batches" improves training, which is consistent with smaller gradient variances. One way to show that curriculum learning is actually at play here would be to do an "anti-curriculum" that starts with long sequence and ends with short sequences.
- It seems to be that it will be hard for curriculum learning to gain traction for pretraining. The cost of pretraining is high, and so introducing additional complexity will probably be done with caution, and if curriculum learning requires additional hyperparameters, people may hestitate to use it in case they do not choose correct curriclum learning hyperparameters.

Minor comments
- Cite the transformer paper in sentence 1
- It is probably not best practice to cite the GPT-3 training time if you aren't doing GPT-3. Just cite the GPT-2 training time instead.
- The top margin seems too small.

**Summary Of The Paper:**

This paper focuses on the pretraining of large language models. They address the issue of instability by proposing curriculum learning. They say that curriculum learning enables training with 8x larger batch size and 4x larger learning rate. This reduces number of tokens and wall clock time by the order of magnitude of 2x.

**Summary Of The Review:**

This paper studies a straightforward pretraining strategy. It does not propose a novel method, and is not well-motivated enough as an empirical evaluation, in my opinion. Given prior work, results were not unexpected. At this point, I cannot clearly understand the paper's claims regarding "stability" and "regularization", and hence cannot recommend acceptance to ICLR (though I think it's fine as a workshop paper).

---

> ### Author Response · Authors · 2021-11-21
> **Rebuttal reply part 1**
>
> Thank you for your comments and below are our replies to some of them. In addition, in the rebuttal paper revision we added some new experiments/analysis related to the comments from reviewers: 1) Appendix A.3 where we perform additional analysis about training stability. 2) Appendix A.4 where we perform additional analysis about learning rates and gradient clipping. 3) Appendix A.5 where we perform additional analysis about sequence length.
>
> <Comment 1> "In terms of new ideas, the paper is not particularly ambitious or novel..."
>
> <Reply 1> We politely argue that our work demonstrates the follow novel contributions: (a) We are the first work to show that curriculum learning enables stable and more efficient training with larger batch sizes and learning rates. Among all the previous curriculum learning works we found, all the works focus on applying curriculum learning with the same training hyperparameters, and only focus on the convergence speed difference; (b) Based on Adam optimizer states, we provide an in-depth analysis of why curriculum learning could improve training stability and convergence speed. Among all the previous curriculum learning works we found, we did not find any similar analysis; (c) In Appendix A.1 we provide a novel and low-cost tuning strategy for curriculum learning which only requires running the very beginning of the training. Among all the previous curriculum learning works we found, most of them only show results based on manually-picked curriculum learning hyperparameters without providing a well-defined tuning strategy. Overall, our work shows great convergence and stability gains, explains why curriculum learning provides such gain, provides a clear and low-cost tuning strategy, and can be easily integrated without nontrivial data processing as in previous works.
>
> <Comment 2> "The paper claims that the new lesson learned is that stability is improved with curriculum learning. If a convincing result for this was shown, I have missed it. I think part of it is that it is still unclear to me what the particular operationalization of "stability" is that is used in the paper. The fact it is really hard to match the colors to lines in Figure 3 doesn't help, either. For instance, where is the training instability in Figure 2? Did the GPT-2 authors originally struggle a lot to do "stable" training? If there is prior work on operationalizing stability, then it should probably be explained further in the paper, given that stability is the novel contribution."
>
> <Reply 2> We agree that we did not do a good enough job to illustrate and quantify the instability. In the rebuttal paper revision we added Appendix A.3 where (a) We re-plotted Figure 2(a) and 3(a) but with less y-axis truncation, to clearly show the training loss spikes that mostly happened at the first half of pre-training; (b) We defined "training instability" by a training loss ratio measurement, and used this definition to quantify training instability and demonstrated that the proposed work greatly reduced the training instability issue. As described in Section 2 and 3, the original GPT-2 work used a small batch size 512 which has less instability issue as we found in our work. The original GPT-2 work didn't mention why a larger batch size is not used, but given the large enough number of GPUs they used, our guess is that they faced the same stability issues. In addition, recently a research project reported on Twitter that they are facing serious training stability issue when trying to pre-train a 100B+ GPT-style model (https://twitter.com/Thom_Wolf/status/1447565680384032776).
>
> <Comment 3> "To make the above bullet point even worse, the use of the word "regularization" further convolutes what the main learning is..."
>
> <Reply 3> Based on wikipedia definition, overfitting is "the production of an analysis that corresponds too closely or exactly to a particular set of data, and may therefore fail to fit additional data or predict future observations reliably". It is true that in machine learning area overfitting is usually referring to the condition where the model overfit on the whole training data which leads to worse or unstable performance on validation data. However, in our case a slightly different kind of overfitting did happen at the beginning of training: Because of the large model size and the very small number of sequences that the model sees at the beginning of training, it is very easy for the model to overfit on the early training data, which leads to worse and unstable performance on later training data. Thus we believe that the proposed work can be regarded as a kind of regularization: curriculum learning regularizes the early training to only learn from simpler examples, to prevent the overfitting which hurts the learning on later training data. On the other hand, we agree that additional clarification might be needed in the paper about this topic, and we would like to hear your opinion on this.

---

> ### Author Response · Authors · 2021-11-21
> **Rebuttal reply part 2**
>
> <Comment 4> "The paper would be more attractive in my opinion if they really made it more about curriculum learning than it currently is. What I mean by this is that the current paper seems more like "BERT trains fast and with more stability if you gradually increase the sequence length of examples." It would be a lot more interesting, for example, if it were shown that the author's proposed method worked for other operationalizations of curriculum learning in addition to sequence length (e.g., age of acquisition, word frequency, etc)."
>
> <Reply 4> We totally agree that it would be interesting to explore different curriculum difficulty metrics. However, (a) As described in Section 2 only a few works applied curriculum learning to large-scale pre-training tasks. These works use staged curriculum learning based on sequence length only. And among all the curriculum learning literature that we found in the broader NLP area (without evaluation on pre-training tasks), we did not find a single curriculum metric proved to substantially outperform other metrics; (b) As described in Section 4, length-based metric has unique values including performance gain for transformer models and zero additional data processing cost based on our solution. Both are important for large-scale pre-training with a large training data corpus, and the data processing cost actually is a challenge we are still trying to solve.
>
> <Comment 5> "Related to the above point, there is very little motivation for why sequence length is a correct operationalization."
>
> <Reply 5>  As described in Section 2, sequence length is used in many previous works as a kind of curriculum learning in NLP area. Shorter sequences are not necessarily easier examples, but shorter sequences are definitely simpler examples since there are less context to embed so that there is less chance for the model to learn from a "noisy gradient" especially during the early stage of training. In the original curriculum learning paper by Bengio et al, simpler examples are regarded as curriculum learning:"Initially, the weights favor "easier" examples, or examples illustrating the simplest concepts, that can be learned most easily.".
>
> <Comment 6> "In fact, I do not think it is even shown that curriculum learning is the reason for the improvement. For example, the author's experiments do not distinguish curriculum learning from the simple hypothesis "putting similar examples into batches" improves training, which is consistent with smaller gradient variances. One way to show that curriculum learning is actually at play here would be to do an "anti-curriculum" that starts with long sequence and ends with short sequences."
>
> <Reply 6> First, we believe anti-curriculum will not work since (a) anti-curriculum will start from the full sequence length; (b) baseline in our paper already shows that starting from full sequence length would lead to training instability at the beginning of training. Second, we added an experiment in the rebuttal paper revision Appendix A.5, which shows that adding 10 percent of full-length batches to curriculum learning would incur instability spikes similar to baseline. This demonstrates that curriculum learning based on sequence length is the reason for the improvement. Regarding "putting similar examples into batches", we agree that it might be able to provide similar stability gain as curriculum learning. However, such technique (a) is orthogonal to curriculum learning since we can both group similar data and sort them by difficulty; (b) grouping similar training data would be non-trivial for pre-training, since pre-training data are webtext corpus that are huge, less-processed, and noisy.

---

> ### Author Response · Authors · 2021-11-21
> **Rebuttal reply part 3**
>
> <Comment 7> "It seems to be that it will be hard for curriculum learning to gain traction for pretraining. The cost of pretraining is high, and so introducing additional complexity will probably be done with caution, and if curriculum learning requires additional hyperparameters, people may hestitate to use it in case they do not choose correct curriclum learning hyperparameters."
>
> <Reply 7> We agree that your concerns apply to many previous works about curriculum learning which could be hard to tune and use. However, as described in Section 4, our approach is easy to integrate, without the need of preprocessing data as in traditional curriculum learning implementations. As described in Appendix A.1, we provide a novel and low-cost tuning strategy for curriculum learning which only requires running the very beginning of the training. As described in Appendix A.4 added in the rebuttal paper revision, our approach also reduces the necessary tuning effort on normal training hyperparameters like learning rate, since the proposed work enables stable training on a wider range of hyperparameter choices. Because of this ease of integration and tuning, our work actually has already been adopted by users inside and outside our organization (cannot provide details which may violate the anonymity).
>
> <Comment 8> "Cite the transformer paper in sentence 1"
>
> <Reply 8> We have applied the suggestion in the rebuttal paper revision.
>
> <Comment 9> "It is probably not best practice to cite the GPT-3 training time if you aren't doing GPT-3. Just cite the GPT-2 training time instead."
>
> <Reply 9> We have added the GPT-2 training time in the rebuttal paper revision. On the other hand, we kept the GPT-3 training time because (a) The training time is helpful to motivate the increasing cost of pre-training tasks; (b) Because of the similar model architecture, our proposed work would be beneficial for all GPT-style models.
>
> <Comment 10> "The top margin seems too small."
>
> <Reply 10> Thank you for pointing this out. We found that we mistakenly used the fullpage package which reduced the top margin. We fixed it in the rebuttal paper revision, and after the fix our paper (including the original submission) still meets the page limit.

---

> ### Comment · Reviewer_D2sE · 2021-11-22
> **Some concerns alleviated, but still not excited about the paper**
>
> Replies 2, 3, 7 seem good. Maybe I did not appreciate the difficulty of training these very large neural nets enough, so the authors might make their paper more appealing to a general audience if they described these a bit more in the paper.
>
> At the end of the day, I am not super excited about the paper because most of what is shown (sort by sequence length and then train) is not particularly new, and the results are not particularly surprising.
>
> I was really curious about the anti-curriculum response, as I do not understand how the author's response solves the issue. Regarding the first reason the author mentioned, if the hypothesis is that the instability comes from too diverse of a range of data (diverse means wide range of sequence lengths), then this does not explain why the anti-curriculum would not work. The second experiment also would not show this, because again there is a diverse range of data (adding 10% full length), so I am not sure that this is adequately responded to. Maybe a good way to do this would be to actually do the experiment, and you can stop early when the training totally fails.
>
> On comment 7, I am halfway convinced by the fact other people are using this, but maybe it would be great to show that the exact same technique extrapolates to other pretraining settings (do you already have this?). If the exact same strategy works for many settings without any hyperparameter tuning, then I think this would be much more useful to recommend to colleagues. But I think one of the reasons why curriculum learning has not become mainstream is that it mostly does not work for any given setting? (Correct me if I'm wrong)
>
> I also feel bad that the authors did such expensive experiments and will not get a paper out of it. I will raise my score from 3 to 5.

---

> > ### Author Response · Authors · 2021-11-22
> > **Reply**
> >
> > Thank you for the reply. Regarding anti-curriculum, we think there might be some misunderstanding about the data sequence length. As described in Section 4 "During the baseline GPT-2 pre-training, the raw text inputs are truncated into sequences with the same length to form a mini-batch regardless of the lengths of the actual sentences". We agree that diversity of sequence length could be a source of instability. However, baseline GPT-2 training data all have the same length 1K. Under this setting, our hypothesis is that longer sequence length leads to longer context information that needs to be captured by self-attention, which is more difficult than short sequences from a curriculum perspective. And the same length sequence is why we cannot sort but actually truncate the sequences. An anti-curriculum for GPT pre-training corresponds to training with long sequences in the beginning, which is basically the same beginning as baseline (given there is no diversity). Baseline has instability issues from the beginning, thus we believe that anti-curriculum will not work. In later version of this paper (not enough time to catch rebuttal deadline) we would add this experiment if resource permits.
> >
> > Regarding curriculum learning for other pre-training settings, we agree and we are exploring some related directions but it is still in progress.

---

### Official Review · Reviewer_W3V1 · 2021-10-17

**Correctness:** 4
**Technical Novelty And Significance:** 2
**Empirical Novelty And Significance:** 2
**Recommendation:** 5
**Confidence:** 5

**Main Review:**

Strengths:
+ The paper is generally well written and easy to follow.
+ The authors provide suficient technical details to reproduce results.
+ The experimental results show the utility of curriculum learning.

Weaknesses:
- The paper applies and existing curriculum learning approach on GPT-2. Hence, the paper lacks novelty. The study could have been performed by other researchers without too much effort. While the work might be suitable perhaps as a workshop, the level of novelty for ICLR is definitely not met.

- It is not clear that truncating text data leads to easier examples. The truncated part might contain relevant features and by removing them, it would be impossible for the model (and for a human) to make a prediction. Therefore, proposed method might also be regarded as anti-curriculum approach. Is there any linguistic motivation that could support truncating the examples?

- The curriculum baselines chosen in Sec. 5.4 are very weak. There are plenty of curriculum learning methods that could have been considered as baselines.

Minor language corrections:
"which helps improves" => "which helps to improve" or "which improves";

**Summary Of The Paper:**

The authors present a curriculum learning approach that gradually increases the input sequence length for training large-scale transformer models. The study various aspects of how the approach influences the training of GPT-2, for example showing that curriculum learning can lead to reduced wall-clock time and number of tokens.

**Summary Of The Review:**

In my opinion, the weaknesses outweigh the strengths of this paper. In particular, the lack of novelty is a major issue that is enough to justify rejecting the paper.

---

> ### Author Response · Authors · 2021-11-21
> **Rebuttal reply**
>
> Thank you for your comments and below are our replies to some of them. In addition, in the rebuttal paper revision we added some new experiments/analysis related to the comments from reviewers: 1) Appendix A.3 where we perform additional analysis about training stability. 2) Appendix A.4 where we perform additional analysis about learning rates and gradient clipping. 3) Appendix A.5 where we perform additional analysis about sequence length.
>
> <Comment 1> "The paper applies and existing curriculum learning approach on GPT-2. Hence, the paper lacks novelty. The study could have been performed by other researchers without too much effort. While the work might be suitable perhaps as a workshop, the level of novelty for ICLR is definitely not met."
>
> <Reply 1> We politely argue that our work demonstrates the follow novel contributions: (a) We are the first work to show that curriculum learning enables stable and more efficient training with larger batch sizes and learning rates. Among all the previous curriculum learning works we found, all the works focus on applying curriculum learning with the same training hyperparameters, and only focus on the convergence speed difference; (b) Based on Adam optimizer states, we provide an in-depth analysis of why curriculum learning could improve training stability and convergence speed. Among all the previous curriculum learning works we found, we did not find any similar analysis; (c) In Appendix A.1 we provide a novel and low-cost tuning strategy for curriculum learning which only requires running the very beginning of the training. Among all the previous curriculum learning works we found, most of them only show results based on manually-picked curriculum learning hyperparameters without providing a well-defined tuning strategy. Overall, our work shows great convergence and stability gains, explains why curriculum learning provides such gain, provides a clear and low-cost tuning strategy, and can be easily integrated without nontrivial data processing as in previous works.
>
> <Comment 2> "It is not clear that truncating text data leads to easier examples. The truncated part might contain relevant features and by removing them, it would be impossible for the model (and for a human) to make a prediction. Therefore, proposed method might also be regarded as anti-curriculum approach. Is there any linguistic motivation that could support truncating the examples?"
>
> <Reply 2> We agree that shorter sequences might not be easier examples, but shorter sequences are definitely simpler examples since there are less context to embed so that there is less chance for the model to learn from a "noisy gradient" especially during the early stage of training. In the original curriculum learning paper by Bengio et al, simpler examples are regarded as curriculum learning: "Initially, the weights favor "easier" examples, or examples illustrating the simplest concepts, that can be learned most easily.". As described in Section 2, sequence length is used in many previous works as a kind of curriculum learning in NLP area.
>
> Regarding truncating text data, GPT-style models only look at the context before the token to predict, so our truncation would not change the context for any token prediction, but only avoid token predictions that have longer contexts. For language models that look at both context before and after the token, we think there are two alternative implementations: (1) Randomly truncate context from both head and tail of the sequence; (2) If the model uses the natural sentences with different lengths as training data (instead of the fixed sequence length used by GPT models), then one can implement curriculum learning in the traditional way that reorders training data based on natural sentence length, so that there will be no truncation.
>
> <Comment 3> "The curriculum baselines chosen in Sec. 5.4 are very weak. There are plenty of curriculum learning methods that could have been considered as baselines."
>
> <Reply 3> We agree that there are plenty of previous works that apply curriculum learning in NLP area. However, as described in Section 2 only a few works applied curriculum learning to large-scale pre-training tasks. These works use staged curriculum learning based on sequence length, and we had compared them in Section 5.4 as related works. Since large-scale pre-training tasks are very different from small-scale one-stage tasks or downstream fine-tuning tasks (e.g., the training stability issue mentioned in our paper), any technique that works on the latter is not guaranteed to work on the former. Thus we politely argue that previous works without pre-training task evaluations cannot be regarded as directly-comparable related works, but we definitely agree that it would be interesting to explore how to apply those methods to pre-training tasks.
>
> <Comment 4> "Minor language corrections..."
>
> <Reply 4> We have applied the suggestion in the rebuttal paper revision.

---

> > ### Comment · Reviewer_W3V1 · 2021-11-21
> > **Reply to rebuttal**
> >
> > I thank the authors for addressing my comments. With respect to the first comment, I still do not find any novelty on the theoretical / methodological side. The claimed novelties are rather related to experiments, and I believe this is not enough for ICLR. Hence, my recommendation stays the same.

---

> > > ### Author Response · Authors · 2021-11-21
> > > **Reply**
> > >
> > > We kindly argue that the novelty of a paper should not be judged solely on the theoretical/methodological side. In particular, this year ICLR review guidelines explicitly mentioned that "Submissions with significant contributions in either technical aspects or empirical aspects will be given high priority for acceptance.". We therefore ask you to reconsider your position of using that reason to reject our paper.

---

### Official Review · Reviewer_LtMz · 2021-11-02

**Correctness:** 4
**Technical Novelty And Significance:** 3
**Empirical Novelty And Significance:** 3
**Recommendation:** 8
**Confidence:** 3

**Main Review:**

Strengths:
- simple but effective curriculum method
- large savings in time and improvements in quality

Weakness:
- Because of the surprising method simplicity, more analysis would be interesting to add that could shad light on the nuances of interplay between gradients and the CL, and why it helps. Some suggestions:
  * Does CL lead to fewer instance of gradient clipping compared to the baseline?
  * "The largest variance on certain dimensions" is mentioned as a problem in the intro and in the last sentences of sec. 5.3, but no experiment measures it w/ and w/o CL.
  * Gradient similarity between neighbouring batches w/ and w/o CL
  * A simulated experiment: while doing CL, insert a sequence of a few baseline (full-length) batches, -- will you see the "instability spikes" like in baseline learning curves?
- Given that Platanios et al reported good results both with length- and word rarity-based curricula, it would be nice to run a few experiments with the word-rarity difficulty definition.
- Regarding tuning the T hyperparameter: since the validation set exhibits fluctuation for some T, I wonder if it's simply because, for larger T, the curriculum hasn't yet ramped up to the actual lengths of validation data? Esp. since the fluctuation seem to fade away closer to the 10K cutoff. If that is true, could you simply set T to a function of some length statistic of the validation set?

Clarification questions/remarks:
- When trimming the batches to the current length, what happens to the shorter sentences? If they stay in the batch, than, effectively it's sampling from all lengths below the current seqlen_t?
- The two-stage curriculum has actually two spikes (@20K in Fig. 3(f) and later @30K), what is the reason for the 2nd spike if the transition to full-length has already happened?
- Figure 4: if CL60K is preferred here, does it mean it overtakes the validation curves for CL100K and CL80K? When does that happen?
- Some of the prior work is mistakenly described as using fine-tuning (e.g. most references in the paragraph starting with "In the natural language processing area,..."), while they are actually using one-stage training, without fine-tuning, as is standard in machine translation.

Minor remarks:
- "extremely huge" -> "extremely large" or simply "huge"
- "the gradient variance norm" -> "the norm of this variance" (to be more specific)
- "Inspired of the highly organized curriculum": what does the definite article refer to? probably, can omit it
- "human and animal," -> "humans and animals,"
- could you add a citation to the sentence ending with ", and model divergence." in sec. 2?
- something is wrong with grammar in the sentence "To quantitatively measure the token...", sec 5.1
- Table 2: I'd suggest using more intuitive '+' and '-' instead of arrows-up and arrows-down
- Tables 1 and 2 are hard to parse because they contain different types of information: consider splitting into 2 half-tables horizontally in the middle (i.e. on table for target ppl and the rest in the other)
- a coma missing after "Because of 1)"
- please use more specific wording instead of "we find that this is not ideal" (both in sec. 5 and A.2). What do you mean - not ideal for 117M or in general? What is "ideal" here?
- Figure 2. "the first 60K" -> "the first T=60K"
- please explain the term "token reduction" in sec 5.4

**Summary Of The Paper:**

The paper proposed a length-based curriculum to pretrain GPT models, that linearly increases the input lengths until a maximum value over the first 20-100K steps. Such curriculum are often proposed and evaluated for different NLP tasks (e.g. Platanios et al for MT), however, they seem to have not been evaluated for GPT models. Despite its simplicity, the approach achieves more stable training, faster convergence to a target perplexity (⪆50% time reduction), and better generalization as measured by validation perplexity or downstream accuracy. The method has three additional hyperparams (initial/final sequence lengths and the curriculum duration in updates), which are relatively easy to set by observing initial model behaviour on the validation set.



**Summary Of The Review:**

A simple approach that nevertheless fixes GPT's stability issues and improves empirical performance.

---

> ### Author Response · Authors · 2021-11-21
> **Rebuttal reply part 1**
>
> Thank you for your comments and below are our replies to some of them. In addition, in the rebuttal paper revision we added some new experiments/analysis related to the comments from reviewers: 1) Appendix A.3 where we perform additional analysis about training stability. 2) Appendix A.4 where we perform additional analysis about learning rates and gradient clipping. 3) Appendix A.5 where we perform additional analysis about sequence length.
>
> <Comment 1> "Because of the surprising method simplicity, more analysis would be interesting to add that could shad light on the nuances of interplay between gradients and the CL, and why it helps. Some suggestions: Does CL lead to fewer instance of gradient clipping compared to the baseline?"
>
> <Reply 1> We agree and we appreciate your suggestions of what additional analysis would be interesting to add. For the number of gradient clipping, in the rebuttal paper revision we added Appendix A.4.2 where we perform additional analysis about gradient clipping, and found that CL leads to 21 percent fewer gradient clipping.
>
> <Comment 2> ""The largest variance on certain dimensions" is mentioned as a problem in the intro and in the last sentences of sec. 5.3, but no experiment measures it w/ and w/o CL."
>
> <Reply 2> Actually we did measure and plot the max dimension-wise variance at each step in Figure 3(f) in terms of Adam variance max element, and results show that our works avoids extreme max elements that occur for baseline and related works. It is true that we did not log the max value on every specific dimension (maybe "The largest variance on certain dimensions" leads to some confusion), but we think this plotted max element is sufficient as an indicator of the instability risk at each step.
>
> <Comment 3> "Gradient similarity between neighbouring batches w/ and w/o CL"
>
> <Reply 3> We believe that the Adam variance norm plotted in Figure 3(e) is a good approximation of the gradient similarity between neighbouring batches. We set beta2 at 0.999, which means that the variance norm would be smaller if the neighbouring 1000 batches have higher gradient similarity. Thus Figure 3(e) indicates that at batch size 4K CL has higher gradient similarity than baseline. For batch size 512, CL has higher gradient similarity during the first 5K steps where the training has highest instability risk, and after that baseline and CL have similar level of gradient similarity.
>
> <Comment 4> "A simulated experiment: while doing CL, insert a sequence of a few baseline (full-length) batches, -- will you see the "instability spikes" like in baseline learning curves?"
>
> <Reply 4> We added this experiment in the rebuttal paper revision Appendix A.5, and found that adding 10 percent of full-length batches to CL does incur instability spikes similar to baseline, plus some other interesting findings.
>
> <Comment 5> "Given that Platanios et al reported good results both with length- and word rarity-based curricula, it would be nice to run a few experiments with the word-rarity difficulty definition."
>
> <Reply 5> We totally agree that it would be interesting to explore different curriculum difficulty metrics, which have not been explored for pre-training. However, (a) In Platanios et al the length and word rarity metrics provide quite similar time savings (no more than 1.3x difference). And among all the CL literature that we found, we did not find a single curriculum metric proved to substantially outperform other metrics; (b) As described in Section 4, length-based metric has unique values including performance gain for transformer models and zero additional data processing cost based on our solution. Both are important for large-scale pre-training with a large training data corpus, and the data processing cost actually is a challenge we are still trying to solve.

---

> ### Author Response · Authors · 2021-11-21
> **Rebuttal reply part 2**
>
> <Comment 6> "Regarding tuning the T hyperparameter: since the validation set exhibits fluctuation for some T, I wonder if it's simply because, for larger T, the curriculum hasn't yet ramped up to the actual lengths of validation data? Esp. since the fluctuation seem to fade away closer to the 10K cutoff. If that is true, could you simply set T to a function of some length statistic of the validation set?"
>
> <Comment 7> "When trimming the batches to the current length, what happens to the shorter sentences? If they stay in the batch, than, effectively it's sampling from all lengths below the current seqlen\_t?"
>
> <Reply 6 and 7> First, we think there might be some misunderstanding about the data sequence length. As described in Section 4 "During the baseline GPT-2 pre-training, the raw text inputs are truncated into sequences with the same length to form a mini-batch regardless of the lengths of the actual sentences". So for baseline all the sequences have exact the same length 1K. For CL, we truncate the length of training data but keep the validation/test data at full-length to have a fair validation comparison between baseline and CL. Regarding comment 6, it is true that with larger T CL will see more gaps between the training and validation seqlen at the beginning. For too large T, staying at short training seqlen could lead to the model overfitting to those shorter training data, which leads to the validation fluctuation and we found that this also indicates a worse test PPL/zero-shot eval at the end of pre-training. So we think that T is not related to the length of validation data, but is related to the eval perplexity on full-length validation data.
>
> <Comment 8> "The two-stage curriculum has actually two spikes (@20K in Fig. 3(f) and later @30K), what is the reason for the 2nd spike if the transition to full-length has already happened?"
>
> <Reply 8> We believe that this is because in general staged CL approaches cannot provide the same level of stability gain as the proposed linear function approach. Our experience is that a smoother seqlen function provides long lasting stable training effect. For the two-stage CL case, the sudden transition from sequence length 128 to 1K can cause a large variance in model weights and optimizer states that can still lead to training spikes manifested in later steps even after the full-length is reached.
>
> <Comment 9> "Figure 4: if CL60K is preferred here, does it mean it overtakes the validation curves for CL100K and CL80K? When does that happen?"
>
> <Reply 9> Yes, in the rebuttal paper revision we added Figure 4(b) and some description in Appendix A.1 to show that CL 60K provides the best validation curve in the second half of the training.
>
> <Comment 10> "Some of the prior work is mistakenly described as using fine-tuning (e.g. most references in the paragraph starting with "In the natural language processing area,..."), while they are actually using one-stage training, without fine-tuning, as is standard in machine translation."
>
> <Reply 10> We agree and in the rebuttal paper revision we changed it to "small-scale one-stage tasks and downstream fine-tuning tasks".
>
> <Comment 11> "Minor remarks: ..."
>
> <Reply 11> Thank you and in the rebuttal paper revision we directly applied most of your suggestions.
>
> For "Tables 1 and 2 are hard to parse because ..." do you actually mean tables 1 and 3? Unfortunately we didn't find a way to split the tables while still meeting the page limit. As a potential solution we modified horizontal line styles and extended caption to emphasize the difference between first and second two rows.
>
> For "we find that this is not deal" we changed it to "we find that simply increasing decay steps still leads to sub-optimal learning rate schedule".
>
> For "token reduction" we changed the sentence to "Although both batch size warmup and curriculum learning reduce the number of tokens per batch in a similar fashion, batch size warmup does not provide any training stability benefit compared to the baseline."

---

> > ### Comment · Reviewer_LtMz · 2021-11-22
> > **UPD**
> >
> > Most of authors' replies are satisfactory, and I appreciate the effort in adding experiments. The paper has moved beyond the score 6 (sets a new bar for curricula in data/model sizes and in analysis) and would have landed squarely on 8 ("accept, good paper") if other dimensions of complexities would be evaluated which is also requested by reviewer D2sE. (Engineering-wise it's awesome that it works for length, which comes for free, but research progress-wise it doesn't inform us why nor whether there are better complexities.) For the lack of the intermediate level 7, which I would have chosen, I'm bumping my recommendation score to 8.

---

### Official Review · Reviewer_qUd8 · 2021-11-03

**Correctness:** 2
**Technical Novelty And Significance:** 3
**Empirical Novelty And Significance:** 3
**Recommendation:** 5
**Confidence:** 4

**Main Review:**

Training massive transformer models in a stable manner has emerged as a challenge for practitioners trying to take advantage of the benefits of model scaling. As such, providing new tools to stabilize the training dynamics is of significant interest to the community. The proposed paper indeed demonstrates significant improvements in training stability and speed.

This being said, there are a number of issues that the paper does not properly address:

1- How general are the results of the paper? Do we observe similar instability issues for other causal transformers beyond GPT-2? If so, is the proposed curriculum learning approach effective there?

2- The experimental design is not clear: The paper compares only two different hyper-parameter choices. These two hyper parameter choices differ in every aspect (learning rate, batch size, #steps). As such, it is unclear where the instability issue is coming from. Moreover, from the paper, it is unclear to me if these instability issues represent fundamental training limitations that require an involved solution as presented in the paper. It might be the case that better tuning of the hyper-parameters, alongside with more careful clipping of the gradients and activations is just enough to stabilize training.


**Summary Of The Paper:**

The paper examine optimization stabilities encountered during training of GPT-2 models. It demonstrates that as model size, learning rate, and batch sizes grow training dynamics of the model become unstable. To alleviate these issues, the authors propose a curriculum learning approach based on the length of the sequence. They show that this approach stabilizes the learning dynamics and drastically improves the training efficiency.

**Summary Of The Review:**

While the paper describes an intriguing phenomenon, the training setups the paper uses as baselines are exceedingly weak and underdeveloped. As such, it is not clear to me that the paper is presenting a real improvement to our training setup.

---

> ### Author Response · Authors · 2021-11-21
> **Rebuttal reply**
>
> Thank you for your comments and below are our replies to some of them. In addition, in the rebuttal paper revision we added some new experiments/analysis related to the comments from reviewers: 1) Appendix A.3 where we perform additional analysis about training stability. 2) Appendix A.4 where we perform additional analysis about learning rates and gradient clipping. 3) Appendix A.5 where we perform additional analysis about sequence length.
>
> <Comment 1> "How general are the results of the paper? Do we observe similar instability issues for other causal transformers beyond GPT-2? If so, is the proposed curriculum learning approach effective there?"
>
> <Reply 1> In terms of instability issue, previous works (e.g., Platanios et al. 2019 in our paper reference, and Popel et al. 2018 "Training Tips for the Transformer Model") found that transformer models in general is less stable during training. So we agree that it would be interesting to explore similar curriculum learning techniques on other transformer models (especially for pre-training large models) that may have similar instability issue. It'd be great if you could let us know which model in particular that you are referring to.
>
> On the other hand, we would want to emphasize the value of enabling more stable and efficient GPT model pre-training: (a) Because of the similar model architecture, our proposed work would be beneficial for all GPT-style models; (b) GPT-style models currently have the largest number of parameters among pre-trained models (excluding MoE models which require less training cost), including 175B GPT-3 and recently announced Megatron-Turing NLG 530B model. Thus it's reasonable to assume that GPT model pre-training is the most computation-heavy training in the world, and our work's 3.3x time saving (that can be easily integrated/tuned and compatible with many other techniques) to GPT pre-training would be a substantial time/energy saving to the world.
>
> <Comment 2> "The experimental design is not clear: The paper compares only two different hyper-parameter choices. These two hyper parameter choices differ in every aspect (learning rate, batch size, \#steps). As such, it is unclear where the instability issue is coming from. Moreover, from the paper, it is unclear to me if these instability issues represent fundamental training limitations that require an involved solution as presented in the paper. It might be the case that better tuning of the hyper-parameters, alongside with more careful clipping of the gradients and activations is just enough to stabilize training."
>
> <Reply 2> We agree that changing multiple hyperparameters at the same time makes it less clear where the instability issue is coming from, and that more tuning experiments would be helpful to demonstrate the necessity of curriculum learning. In the original submission we provided a low-cost tuning strategy for curriculum learning's own hyperparameters in Appendix A.1. In the rebuttal paper revision we added Appendix A.4 where we perform additional analysis about learning rates and gradient clipping. The short summary is that (a) Compared to baseline, curriculum learning requires less tuning effort on these hyperparameters to provide a stable training; (b) By enabling stable training on larger learning rates, curriculum learning could provide better training efficiency and convergence (as demonstrated in main paper Section 5); (c) Tuning gradient clipping for baseline could not provide the same training stability as curriculum learning.

---

> > ### Comment · Reviewer_qUd8 · 2021-11-27
> > **Reply to Rebuttal**
> >
> > I thank the authors for adding Appendix A.4. Indeed these experiments are a step in the right direction. However, there are still certain aspects of the research process missing in the paper, which (I believe) is why some of the other reviewers and I are not rating the paper higher.
> > I believe that a satisfactory answer to the instability phenomenon the authors observe should discuss
> > 1. What causes the training instabilities (spikes). Appendix A.5 briefly suggests that the spikes might be attributed to the longer sequences. This analysis should be expanded to describe exactly what is happening to the optimization dynamics.
> > 2. Can't we use training stabilization approaches proposed in the previous literature to address the issue? For example, if the issue is caused by outlier observations or rare tokens why can't we use techniques such as data filtering, regularization of the embedding weights, or increasing Adam's epsilon parameter for the embeddings? These are popular techniques heavily used when training large models over noisy corpora.
> > 3. What is the mechanism by which curriculum learning (using sequence length) improves the optimization dynamics?
> >
> > Without clear answers to these three questions, the contributions of the paper are hard to assess. From reading the current manuscript, it is unclear to me if the paper is addressing a fundamental issue in training GPT-style models: Training large models (particularly via complex training pipelines such as Megatron) can become unstable due to a huge range of bugs / poor hyper-parameter choices; if the proposed curriculum learning approach is just indirectly alleviating such superficial issues, then it might not merit publication as a new algorithm. However, if the proposed approach is addressing a fundamental training instability limitation, then (as authors suggest) the paper can make a substantial contributions to the community. Since the current analysis does not completely illuminate this issue, I keep my original score.

---

> > > ### Author Response · Authors · 2021-11-29
> > > **Reply**
> > >
> > > Thank you for your reply and below are our replies to the comments.
> > >
> > > <Comment 1> "What causes the training instabilities (spikes). Appendix A.5 briefly suggests that the spikes might be attributed to the longer sequences. This analysis should be expanded to describe exactly what is happening to the optimization dynamics.", "What is the mechanism by which curriculum learning (using sequence length) improves the optimization dynamics?"
> > >
> > > <Reply 1> In Figure 3 and 8 we analyzed the optimization dynamics and demonstrated that training instabilities are correlated with spikes in Adam variance norm and max element. Experiments in Appendix A.5 have the same phenomenon and we will add the corresponding figures to the manuscript. On the other hand, we would like to ask what kind of optimization dynamics do you mean specifically? Because the large-scale training instabilities issue is a challenging and ongoing research area with very few publications, we were not able to find any existing optimization dynamics analysis. It is true that our variance analysis might not be sufficient to completely demystify the root cause of instability, but we would like to emphasize that our experiments clearly prove that there are instability issues and the proposed curriculum learning method greatly helps on solving this issue: completely avoids training loss spikes as shown in Table 5, and achieves better zero-shot eval results with up to 2.2x fewer tokens and 3.3x less time as shown in Table 2.
> > >
> > > <Comment 2> "Can't we use training stabilization approaches proposed in the previous literature to address the issue? For example, if the issue is caused by outlier observations or rare tokens why can't we use techniques such as data filtering, regularization of the embedding weights, or increasing Adam's epsilon parameter for the embeddings? These are popular techniques heavily used when training large models over noisy corpora."
> > >
> > > <Reply 2> We agree that there is value to compare curriculum learning with additional existing techniques. However, we politely argue that the contribution of the proposed work is sufficiently demonstrated by the evaluations in our paper. The proposed work is compatible and shall be combined with the existing approaches above. We are not claiming that the proposed work itself will eliminate instability issues for all training tasks, and none of the existing approaches could make such claim. We are proposing a new training stabilization method that (1) no existing work had found that it (curriculum learning) could greatly help on stabilizing training; (2) the method is easy to tune with low cost (Appendix A.1). In addition, we believe that the following circumstantial evidence can demonstrate the contribution of the proposed work beyond existing approaches: (1) The proposed method could achieve better zero-shot eval results with up to 2.2x fewer tokens and 3.3x less time for GPT-2 pre-training. To the best of our knowledge, none of the existing training stabilization approaches were proved to provide similar benefit; (2) We demonstrated that with curriculum learning it's possible to stabilize the billion-scale GPT-2 pre-training with batch size 4K. This is larger than the batch sizes used by existing large-scale pre-training publications as described in Section 2 and 3: the 11B T5 model used batch size 2K, the billion-scale GPT-2 models (both OpenAI and NVIDIA) used batch size 512, and the 175B GPT-3 model used batch size 1.6K.
> > >
> > > <Comment 3> "Without clear answers to these three questions, the contributions of the paper are hard to assess. From reading the current manuscript, it is unclear to me if the paper is addressing a fundamental issue in training GPT-style models: Training large models (particularly via complex training pipelines such as Megatron) can become unstable due to a huge range of bugs / poor hyper-parameter choices; if the proposed curriculum learning approach is just indirectly alleviating such superficial issues, then it might not merit publication as a new algorithm. However, if the proposed approach is addressing a fundamental training instability limitation, then (as authors suggest) the paper can make a substantial contributions to the community. Since the current analysis does not completely illuminate this issue, I keep my original score."
> > >
> > > <Reply 3> Regarding bugs and hyper-parameter choices, we tried our best to follow existing work and our zero-shot evaluation results is consistent with existing results (Shoeybi et al., 2019). As described in the replies above, we politely argue that results and analysis in our paper are sufficient to demonstrate the novel contribution and great benefit of the proposed work. It is true that demystifying training instability is an ongoing research direction that needs further explorations, but we believe that no matter what the root cause is, the fact that "the issue exists and the proposed work helps solving the issue" is well presented in our paper.

---

### Decision · Program_Chairs · 2022-01-20

**Decision:**

Reject

**Comment:**

This submission proposes a simple way to improve the stability of training GPT-2: Increase the sequence length of examples over the course of training. It is shown that this simple heuristic can result in using larger learning rates, therefore significantly speeding up convergence. Reviewers agreed that this was a simple and effective approach, but shared various concerns about the paper:
- The paper focuses on GPT-2, while stability issues can arise in a much wider range of models. Additional experiments with other models (and ideally other codebases/training setups) would help verify that the proposed method is broadly applicable.
- Better analysis of why using the sequence length as the difficulty metric would be helpful. What other criteria would be possible? Why is sequence length the best?

I would suggest that the authors significantly expand the submission based on the above suggestions and resubmit.